# Political-LLM: Large Language Models in Political Science

## Abstract

Political science is undergoing a significant transition as large language models (LLMs) gain traction in tasks such as election forecasting, policy assessment, and misinformation detection. While LLMs advance political research, they also pose challenges, including but not limited to societal biases (e.g., partisan skew in political sentiment analysis), ethical concerns (e.g., misinformation propagation in automated legislative summarization), and scalability limitations (e.g., inefficiencies in adapting general LLMs for real-time election forecasting). In this work, we—an interdisciplinary team bridging computer science and political science—take an initial step towards systematically understanding how LLMs can be integrated in political science by introducing the principled conceptual framework named Political-LLM. Specifically, our approach begins with a taxonomy that divides **normative political science** (NPS) and **positive political science** (PPS), a method of classification that is deeply rooted in the foundation of classical political science research. By grounding the framework in this perspective, we provide a structured view for organizing previous work, pinpointing critical challenges, and uncovering opportunities to promote both empirical research and responsible applications of LLMs. As a case study, we perform empirical experiments using the ANES benchmark to **evaluate state-of-the-art LLMs** through a voting simulation task, focusing on their abilities to generate relevant political features and expose inherent biases. This study highlights how to employ our principled taxonomy as the guidance of specific research problems in this interdisciplinary field, while also provides an vivid and understandable example for general audience to deepen their comprehension on Political-LLM framework. Finally, we outline **key challenges** and **future directions**, emphasizing domain-specific dataset development, careful attention to issues such as bias and opaque modeling processes, acknowledgment of non-scalability constraints, the value of expert involvement, and the importance of proprietary evaluation criteria that meet the needs of this field. POLITICAL-LLM is intended as a **Guidebook** for researchers seeking to apply Artificial Intelligence in political science with care and impact.

## 1 Introduction

Recent progress in Large Language Models (LLMs) has shown strong potential across diverse fields, including healthcare Wornow et al. (2023); Xu et al. (2024c); Yue et al. (2024), finance Huang et al. (2023a); Wu et al. (2023c); Xie et al. (2024a), scientific discovery Zhang et al. (2024c); Liu et al. (2024a); Nguyen et al. (2024); Edwards et al. (2024); Xin et al. (2024), transportation Da et al. (2024b;a); Zhang et al. (2024b); Li et al. (2024b), and education Kasneci et al. (2023); Pinto et al. (2023); Henkel et al. (2024). These capabilities arise from pre-training on web-scale text corpora Minaee et al. (2024), enabling advanced analysis of complex linguistic patterns Zhao et al. (2023); Linegar et al. (2023). Given these successes, political science, a field centered on studying political systems, behavior, and policymaking Moe (2005); Gao et al. (2022), also has the potential to benefit substantially from LLM-based approaches Rotaru et al. (2024); Rodman (2024). Specifically, traditional political science research focuses mainly on analyzing institutional structures, policy impacts, and political behavior through empirical studies and theoretical frameworks Goodin & Tilly (2008); Sabatier (1991). Therefore, researchers usually rely on strategies with heavy manual analysis (e.g., qualitative coding of legislative debates Laver et al. (2021) and hand-labeling political sentiment in textual data Koltsova & Koltcov (2013)) and/or statistical analysis (e.g., regression-based methods for policy

impact evaluation Skovron & Titiunik (2015) and network analysis for studying political influence Ward et al. (2011)). By contrast, LLMs enable automated, large-scale analysis of political texts Törnberg (2023), including speeches Liu et al. (2023); Xu et al. (2024b), legislative documents Yue et al. (2023); Gesnouin et al. (2024), social media posts Törnberg et al. (2023); Najafi & Varol (2024), and news articles Zhang et al. (2024a); Fang et al. (2024). Such strengthened analytical capabilities have enabled revolutionized ways to study political science in related domains. As an example, Breum et al. (2024) demonstrates that opinion dynamics (e.g., opinion changes towards certain political statements) can be easily simulated with LLM-based agent emulation. This raises the controversial debate of whether LLM agents have the potential to serve as low-cost alternatives for human being participants in political leaning surveys. Moreover, there has also been an emerging interest in applying LLMs in a variety of other political science research topics, such as political behavior Rozado (2024), public opinion Breum et al. (2024), policy formulation Rivera et al. (2024), and election dynamics Gujral et al. (2024). We use a cartoon in Figure 1 to illustrate the impressive impact LLMs have brought to the area of political science.

**Gaps Between LLMs & Political Science.** Despite the notable progress achieved by LLMs, two main gaps pose challenges towards systematically integrating LLMs into political science research Halterman & Keith (2024); Mou et al. (2024b). *First*, from a conceptual perspective, there is no systematic framework or guideline that clarifies the conceptual relationship between LLM-based learning tasks and different mainstreams of political science research. Notably, such a gap can result in misalignment between the strengths of LLMs and the specific need under certain contexts of political science research. For example, while certain political tasks (e.g., political debate

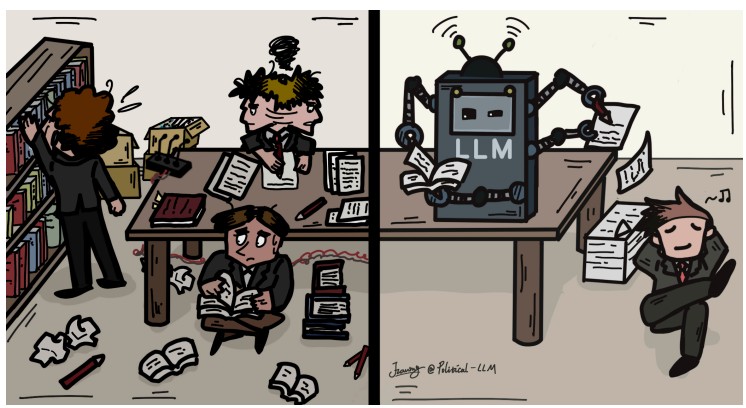

**Figure 1:** LLMs are revolutionizing political science through their advanced language analysis capabilities and the integration of interdisciplinary knowledge.

simulation) require nuanced understanding of context-specific social dynamics, LLMs may default to general language patterns and thus fail to take into account political subtleties (e.g., LLMs may miss strategic ambiguity when analyzing political speeches, such as statements crafted to appeal to multiple voter groups without a firm stance) into consideration Taubenfeld et al. (2024); Cheng et al. (2023). *Second*, from an empirical perspective, current political science studies overwhelmingly rely on general-purpose LLMs while overlooking specific demands such as domain experts' knowledge, the ability to handle ideologically diverse perspectives, and the incorporation of the latest political news into LLM's knowledge database Marino & Giglietto (2024). Although the application of LLMs in political science has gained considerable attention in academia, as reflected by over 300% increase in the number of related publications from 2020 to 2024 Liu et al. (2024b); Kato et al. (2024); Chalkidis (2024), comprehensive discussions on adapting general-purpose LLMs to the research of political science remain limited Linegar et al. (2023); Ziems et al. (2024).

**Political-LLM.** In this paper, we take an initial step to propose a conceptual framework named Political-LLM, which helps researchers and practitioners to integrate LLMs in the political science research under a systematic guidance. Specifically, to tackle the first challenge, this work proposes the first principled taxonomy that connects LLM learning tasks (such as simulating, reasoning, and explaining) with fundamental political science research topics including *positive political science* and *normative political science*. With such a taxonomy, we are able to bridge the LLM computational advancements with traditional political science research. To address the second challenge, Political-LLM advocates domain-adaptive fine-tuning, expert-guided calibration, knowledge augmentation, and ethical standards integration to ensure LLMs align with task-specific requirements, model dynamic political discourse, uphold ethical integrity and transparency, respectively. We validate the contributions through a voting simulation study on benchmarks, demonstrating Political-LLM's role in guiding real-world political applications. The main contributions of this paper are:

- ***Establishing the First Taxonomy for LLM Integration in Political Science.*** We propose the first principled taxonomy bridging LLM learning tasks and fundamental political science research topics. Specifically, this taxonomy systematically connects common LLM learning tasks (e.g., prediction, generation, simulation, explanation) to two of political science mainstream research, namely PPS and NPS. With this taxonomy, we aim to foster interdisciplinary collaborations between researchers in the two fields.

- ***Deriving Key Observations to Guide Specialized LLM Development and Political Applications.*** Our study systematically identifies key observations from the majority of existing research if not all. For PPS, we find that while LLMs excel in text corpora classification, they struggle with explainability, misalignment between LLM technologies and specific political tasks Hristova et al. (2024), and a lack of domain-specific benchmarks. For NPS, we find that ethical risks are amplified in unverified information platforms, and existing bias mitigation techniques require further refinement to fit political contexts.

- ***Exemplary Case Study of LLM Integration into Voting Simulation.*** We conduct a voting simulation case study on the ANES benchmark to demonstrate how Political-LLM framework provides a structured approach to guide task alignment and analyze potential bias. Specifically, we propose to assess how LLMs handle representation imbalances and systemic biases. We further show that structured reasoning enhances predictions on political voting preferences while mitigating ideological distortions.

- ***Characterizing Future Opportunities of Integrating LLMs with Political Science.*** We offer an in-depth characterization of pivotal future research directions. For PPS, we identify opportunities in scaling multilingual and multimodal political tasks, enhancing LLMs for policy simulations, improving explainability, and boosting the reasoning capability for political applications. For NPS, we highlight future explorations in bias mitigation and cultural diversity, ensuring transparency and accountability, and integrating ethical standards for responsible LLM deployment. These insights provide a structured roadmap with interdisciplinary synthesis.

**Difference from Existing Work.** Early studies demonstrated how large-scale political text data can be quantitatively processed and analyzed for political research Wilkerson & Casas (2017); Terechshenko et al. (2020). However, constrained by traditional language models, these works could not capitalize on the transformative potential of LLMs. Recent reviews Chatsiou & Mikhaylov (2020); Rodman (2024) recognize the role of LLMs in advancing various downstream political tasks, but lack systematic frameworks to categorize diverse applications and address nuanced challenges. While specialized surveys Lee et al. (2024b); Argyle et al. (2023b); Linegar et al. (2023); Rozado (2023); Ziems et al. (2024); Ornstein et al. (2022) focus on task-specific discussions, they fall short in proposing actionable methodologies to overcome societal biases, handle multilingual political scenarios, mitigate scalability constraints, or enhance algorithm transparency. Different from the works above, we introduce the first conceptual framework Political-LLM, which provides (1) the first structured taxonomy bridging LLM learning tasks and fundamental political science research topics; (2) key observations from a majority of representative works in this interdisciplinary field; (3) an exemplary case study on the ANES benchmark exemplifies how domain-specific evaluations uncover biases and inform design improvements; (4) future directions that are worthwhile to explore. To show a detailed comparison, we further discuss the major differences between our work and other works in Appendix B.

## 1.1 Literature Collection and Organization Process.

To ensure research transparency, we describe how the literature reviewed in this paper is identified, screened, and categorized.

### 1.1.1 Literature Identification.

We identify relevant literature from two complementary disciplinary perspectives. From the computer science perspective, we focused on studies that apply large language models to political related tasks, including election analysis, policy analysis, political text understanding, bias or fairness assessment, etc. From the political science perspective, we consider works authored by political scientists that provide theoretical, conceptual, or normative insights relevant to the use of language models in political research.

For computer science oriented work, we primarily considered papers published between 2019 and 2025. This period corresponds to the rapid development of modern large language models. A small number of earlier

but influential studies are also included for methodological or historical context. For political science oriented work, we prioritize recent studies when possible but also include classical and widely cited works, given the longstanding theoretical traditions of the field. Literature was identified through major publication venues, well known journals and conferences, and citation based exploration of influential papers.

### 1.1.2 Screening and Selection Criteria.

Identified papers are screened based on topic relevance and conceptual contributions. We include studies that explicitly engage with large language models and political concepts, tasks, or outcomes. We prioritize papers that offer clear empirical patterns, methodological approaches, or normative arguments relevant to Political LLM research. When multiple papers address similar problems, we select representative examples rather than attempt to cover them all. Works that do not directly relate to political science or do not meaningfully involve language models are excluded.

### 1.1.3 Categorization into PPS and NPS.

Each selected paper is categorized as Positive Political Science or Normative Political Science based on its primary research focus and methodological orientation. Papers are classified as PPS when their main contribution is empirical analysis, prediction, or measurement of political phenomena. Papers are classified as NPS when their primary contribution concerns normative evaluation, ethical reasoning, or conceptual reflection on political values. In cases where a paper touches on both dimensions, categorization is determined by the dominant contribution.

## 2 The Framework of Political-LLM

### 2.1 LLMs in Political Science: A Taxonomy

As shown in Figure 2, our proposed taxonomy organizes political research tasks into two main branches: **Positive Political Science** (PPS) and **Normative Political Science** (NPS). PPS focuses on empirical, data-driven tasks, including Predictive Tasks (Section C.1), such as election forecasting and policy impact analysis; Generative Tasks (Section C.2), such as synthesizing political data and modeling voter behaviors; Simulation (Section C.3), leveraging LLM agents to explore complex political interactions and dynamics; Explainability and Causal Inference (Section C.4), applying LLMs to understand causal mechanisms and generate counterfactuals. NPS, on the other hand, addresses ethical and societal considerations, encompassing Ethical Concerns in LLM Development and Deployment (Section C.5), focusing on bias mitigation, fairness, and ethical frameworks, and Societal Impacts (Section C.6), analyzing the influence of LLMs on political campaigns, public communication, and misinformation prevention. This taxonomy intuitively reflects the dual nature of political science research, bridging empirical analysis and value-driven inquiry to provide an integrated framework for the discipline. The top five circular points present our insights on challenges & research outlooks for PPS, while the bottom five highlight present our insights on challenges & research outlooks for NPS. These elements showcase a principled framework, guiding researchers to align task-specific demands with interdisciplinary methodologies while addressing technical, societal, and ethical challenges in applying Political-LLMs. Moreover, it reflects the theoretical evolution of political science, illustrating the enduring significance and interplay between its two branches.

As early as the mid-20th century, Dahl (1961) laid the groundwork for political science taxonomy by recognizing the discipline's fundamental dichotomy: one branch is concerned with "what is" (positive), while the other addresses "what ought to be" (normative), emphasizing that this dual focus enables political science to balance empirical inquiry with ethical imperatives. Building on this foundation, Bergsten (1981) argued that normative theories require empirical grounding to avoid utopianism, emphasizing the necessity of integrating normative ideals with positive validation to ensure practical relevance. Gerring & Yesnowitz (2006) further extended this idea by advocating for a synthesis of normative and positive methods, where normative questions draw on empirical evidence, and empirical research is guided by normative principles. In recent decades, the interplay between positive and normative political science has deepened. Steiner (2012) emphasized that normative theory can draw empirical insights to make ethical debates more actionable and policy-relevant.

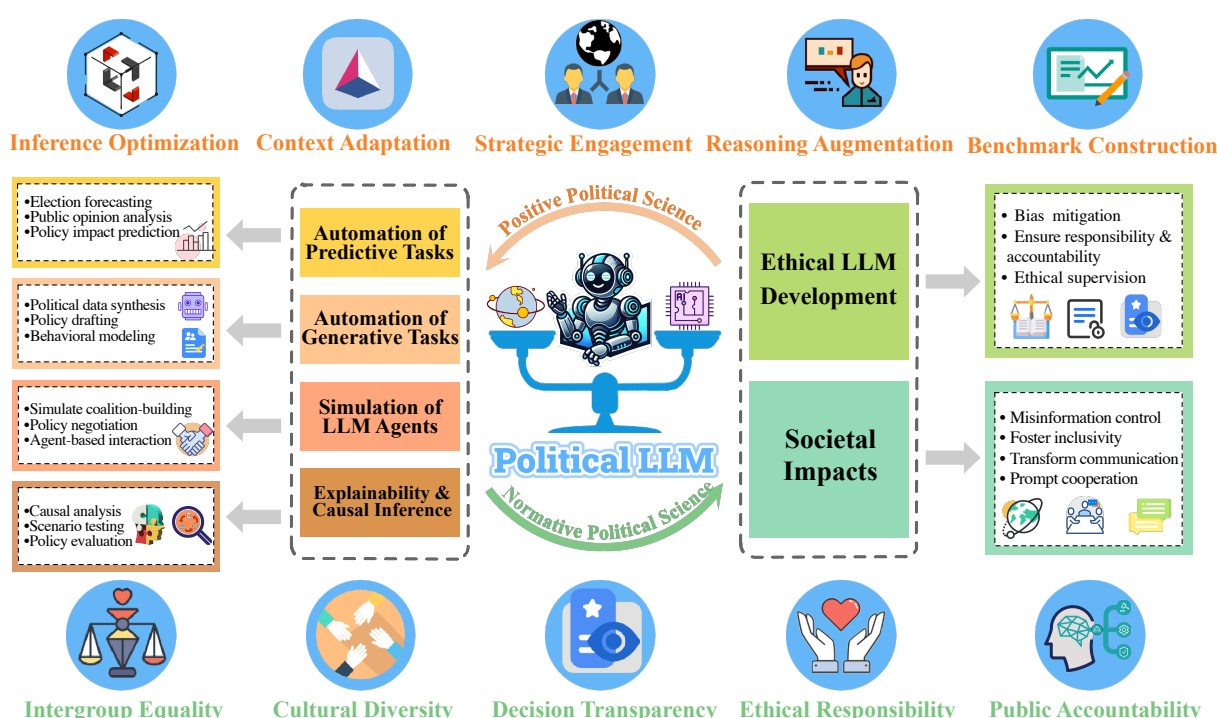

**Figure 2:** The designed architecture framework of POLITICAL-LLM. This framework systematically categorizes the integration of LLMs into political science by distinguishing between Positive Political Science (left, orange) and Normative Political Science (right, green). The top and bottom five icons highlight the future research challenges and opportunities in this interdisciplinary domain.

Most recently, Chung & Kogelmann (2024) demonstrated that formal models (i.e., abstract, mathematical representations of political processes or theories) can bridge the gap between the two branches by rigorously testing normative principles and enabling their application in contexts lacking empirical data.

## 2.2 Practical Structure and Workflow of Political-LLM

To make the Political-LLM framework more operational, we outline its practical workflow, which consists of three sequential stages: problem formulation, model alignment, and evaluation. In the first stage, researchers identify a political research question and determine whether it falls within the scope of Positive Political Science (PPS), which emphasizes empirical prediction, simulation, or measurement; or within Normative Political Science (NPS), which focuses on ethical reasoning, fairness, and accountability. In the second stage, the model alignment process involves selecting or adapting large language models in ways that reflect the theoretical assumptions of the chosen dimension, such as aligning prompt design and data representation with the relevant political context. Finally, in the evaluation stage, researchers assess outcomes using dimension-appropriate criteria. For PPS tasks, the focus is on empirical validity and reproducibility. For NPS tasks, the emphasis is on transparency, bias mitigation, and normative consistency.

This structured workflow provides a clear, step-by-step guide for applying the Political-LLM framework, enabling researchers to translate conceptual principles into concrete analytical or ethical investigations.

## 2.3 Observations of LLMs in PPS Research

Our observations from the perspective of PPS focus on understanding how LLMs perform in quantitatively analyzing those political phenomena of interest with empirical evidence. These observations are drawn from tasks spanning from policy analysis and legal reasoning to voting results prediction and diplomatic strategy modeling. This focus reflects the increasing integration of LLMs into data-driven political research, emphasizing the advancements and limitations of their applicability.

(a) **LLMs Demonstrate Superior Performance in Political Discourse Analysis but Struggle with Policy and Legal Reasoning:** LLMs excel at analyzing political discourse, including classifying political sentiment, detecting ideological biases, and identifying misinformation patterns Huang et al. (2023b); Liu et al. (2025a), with accuracy rates improving by 10%-20% on benchmark datasets compared with specialized models for political science applications Argyle et al. (2023b); Lee et al. (2024b). However, they struggle with policy reasoning, legislative modeling, and impact assessment, which require a deep understanding of institutional constraints, legal structures, and socio-political dynamics. Existing LLMs fail to accurately assess policy feasibility, legislative trade-offs, and causal relationships in governance, with accuracy rates dropping by up to 25% compared to their performance in standard political text classification tasks (e.g., political sentiment classification) Bosley (2024), limiting their effectiveness in policy simulations, legal drafting, and decision-making support.

(b) **Open-Weight LLMs Are Closing the Performance Gap with Proprietary Models in Applications Involving Underrepresented Communities:** Recent experimental evaluations show that open-weight LLMs, such as LLaMA3.2-90B, demonstrate comparable or even superior performance to proprietary models like GPT-4o in political corpora analysis for underrepresented linguistic and demographic groups Kim et al. (2024). Specifically, in non-English political corpora, such as Spanish and German, fine-tuned open-weights LLMs have led to F1-score improvements of 10%-12% Ziems et al. (2024); Pan et al. (2024) compared to proprietary models like GPT-4o. Additionally, for underrepresented groups, open-weight LLMs exhibit stronger adaptability than proprietary models when fine-tuned on domain-specific datasets. This suggests that open-weight LLMs hold significant potential for cost-effective and scalable political research, particularly in diverse linguistic, regional, social, and political settings.

(c) **Scaling LLMs Enhances Generalizability Across Tasks but Risks Misalignment Between LLM Techniques and Specialized Political Applications:** Comparative experiments revealed that larger-scale LLMs (e.g., GPT-5) exhibit stronger generalizability across various political tasks Argyle et al. (2023b); Liu et al. (2025b). In dynamic scenarios such as voter turnout prediction and diplomatic strategy modeling, these models achieve 10%-15% higher accuracy compared to smaller-scale counterparts (e.g., GPT-4o-mini and LLaMA3.1-8B) Lee et al. (2024b). However, as LLMs scale up, their alignment with general-purpose training objectives can lead to misalignment with specialized political tasks. This misalignment arises from over-reliance on broad, non-domain-specific pretraining data, insufficient adaptation to nuanced context, and optimization schemes favoring neutrality over engagement, limiting their ability to generate informed responses to complex political queries Rozado (2023). The findings highlight the trade-off between generalization and domain-specific adaptability, underscoring the need for task-aware fine-tuning and context-sensitive adaptation in LLM development.

(d) **Domain-Specific Benchmarks Remain Scarce Despite the Increasing Adoption of LLMs in Political Science:** Despite the development of proprietary political datasets and political ideology benchmarks very recently, less than 25% of the research we investigated actively used domain-specific benchmarks for LLM evaluation, compared to more than 75% relying on general-purpose datasets for experiments Agiza et al. (2024); Xu & Li (2024). The mismatch restricts the accuracy and comparability of research findings in specialized tasks such as election forecasting or legislative analysis Yu et al. (2024a). The broader adoption of political benchmarks and task-specific metrics remains a recurring challenge.

## 2.4 Observations of LLMs in NPS Research

Our observations from the perspective of NPS highlight the ethical, equity, and accountability challenges closely associated with the use of LLMs in politically sensitive contexts. These include the amplification of misinformation that threatens democratic integrity, reinforcement of structural inequities due to algorithmic bias, and the lack of explainability and responsibility in LLM-driven political decision-making processes. Unlike PPS, which focuses on empirical analysis, NPS concerns itself with the broader societal and normative impact of LLMs, emphasizing the urgent need for transparency, bias mitigation, and accountability mechanisms to ensure responsible applications.

(a) **LLMs Can Amplify Political Misinformation and Undermines Democratic Integrity:** LLMs have been increasingly utilized to analyze, simulate, and even generate political discourse, yet LLMs remain highly susceptible to amplifying misinformation and inaccuracies in politically sensitive texts Yang et al.

(2024). Biases in training corpora and lack of contextual awareness can lead to misleading narratives that influence public opinion. Even after mitigation techniques such as reinforcement learning from human feedback (RLHF) are applied, politically charged misinformation detection remains unreliable and inaccurate Rozado (2023); Weidinger et al. (2021). For example, GPT-4 achieves 84% accuracy on neutral datasets but drops to 68% when analyzing politically motivated misinformation campaigns Huang (2024); Vergho et al. (2024). These findings underscore the pressing need to develop enhanced fact-checking strategies for political contexts.

**(b)  Algorithmic Bias in Political LLMs Reinforces Societal Inequity:** Bias exhibited by LLMs in political science often reflects systemic biases in historical and contemporary discourse, which can reinforce societal inequities. LLMs trained on political corpora exhibit biases in race, gender, and ideology, skewing content toward dominant narratives while marginalizing alternatives Rozado (2023); Ornstein et al. (2022). Mitigation strategies like partitioned contrastive gradient unlearning Yu et al. (2023), stereotype content models Omrani et al. (2023), and adversarial debiasing improve sentiment classification and policy analysis by 15%-20% Rozado (2023); Ornstein et al. (2022). Nevertheless, multilingual bias detection remains inconsistent, with error rates exceeding 20% Azizov et al. (2023); Weidinger et al. (2021). More advanced bias correction methods, such as embedding transformations Han et al. (2024) and human-in-the-loop frameworks, are needed to mitigate the potential inequities.

**(c)  Lack of Explainability and Accountability in Political LLMs Hinders Their Responsible Deployment:** Despite their growing role in political forecasting and policy analysis, LLMs usually function as "black-box" systems, making their decision-making opaque and challenging to verify, particularly when generating ideologically charged or legally sensitive content Weidinger et al. (2021). Accountability crises arise when LLMs generate misleading election forecasts, policy analyses, or legislative summaries with unverifiable reasoning Rozado (2023). To address the issue, structural causal modeling and retrieval-augmented generation (RAG) have been proposed to enhance transparency and factual grounding Huang (2024); Vergho et al. (2024). As LLM-driven political systems expand, establishing robust explainability protocols and accountability principles will be crucial for their responsible deployment.

# 3  Case Study: Political Bias and Feature Generation in Voting Simulations

The case study represents a focused example within the broader landscape of LLM applications in political science, illustrating both the potential and challenges of LLM-driven analysis. Among several available tasks and benchmarks (Section D.1), we select voting simulation task using the 2016 ANES benchmark Studies (2019) due to its relevance to key issues in Positive and Normative Political Science, as well as the volume and comprehensiveness of the data. ANES's rich demographic and ideological attributes allow us to explore biases in LLM outputs (Section C.5) and assess their generative capabilities for feature extraction, aligning with our taxonomy established earlier in this work. By examining the case, we demonstrate how such simulations reveal empirical challenges, such as bias quantification and feature quality evaluation, while contributing to advancing methodologies for reliable and unbiased feature generation in politically sensitive contexts. This dual focus provides a robust foundation for analyzing Political-LLM related research.

## 3.1  LLM Configuration and Computational Resources

The case study aims to evaluate *two key aspects*: (1) the quantitative results given by different LLMs to perform voting simulation Qi et al. (2024); Qu & Wang (2024); and (2) the ideological inclination of LLMs when fed with demographic information. To facilitate a more holistic view, we conduct the study from both PPS and NPS perspectives and gain insights on both sides. The ratio of party affiliation (Republicans to Democrats) is predicted using samples from the 2016 ANES dataset, with each observation representing a voter's demographic and ideological labels. This setup ensures simulated voting distributions align with ANES-represented demographic and political tendencies.

We select the following popular general-purpose LLMs as the benchmark models here: (I) gpt4o-mini-base, (II) gpt4o-base, (III) llama3.1-8B-gen, (IV) gpt4o-mini-gen, (V)gpt4o-gen, (VI) llama3.1-70B-base, (VII) gpt4o-NP, (VIII) llama3.1-8B-base, (IX) llama3.1-70B-gen. The hardware configurations were tailored to meet the computational requirements of each model. For GPT-4o and GPT-4o-mini, experiments were conducted on a GPU server equipped with an AMD EPYC Milan 7763 processor, 1 TB of DDR4 memory, 15 TB SSD

storage, and 6 NVIDIA RTX A6000 GPUs. For Llama 3.1 models, a node with 8 NVIDIA A100 GPUs (each with 40 GB of memory), dual AMD Milan CPUs, 2 TB of RAM, and 1.5 TB of local storage was utilized.

### 3.2 Experimental Design

**Experimental Design.** To investigate voting simulation bias and feature generation quality, we adapted methodologies proposed in previous studies Yu et al. (2024b); Arg (2023). In our experimental design, each selected LLM is provided with detailed persona information, including demographic characteristics, political ideology, and religious affiliation, as well as contextual information on candidates and policies relevant to the election year. Each LLM is then employed to simulate election voting behavior for each persona, allowing us to observe any biases that emerged in the simulated vote distributions.

We designed two experimental pipeline setups for each LLM: a baseline group (denoted as [model name]-base) and a generation group (denoted as [model name]-gen). In the base group, LLMs used the original, unaltered ANES dataset inputs to simulate voting behaviors. In the generation group, we applied a multi-step Chain of Thought (CoT) approach, with a specific CoT prompt design illustrated below:

---

**Case study - CoT Prompts example**

**Step 1: Ideology Assessment.** You are a persona with the following demographic characteristics: [demographics]. The current year is [year]. Here are the policy agendas of the two parties:
[Two parties' policy agenda].
When it comes to politics, would you describe yourself as:

  No answer & Very liberal
  Somewhat liberal & Closer to liberal
  Moderate & Closer to conservative
  Somewhat conservative & Very conservative

**Step 2: Voting Simulation.** You are a persona with the following demographic characteristics: [demographics]. Your political ideology is described as [conservative-liberal spectrum]. The current year is [year]. Here are the policy agendas of the two parties: [Two parties' policy agenda]. Additionally, here are the presidential candidates' biographical and professional backgrounds:
[Presidential candidates' biographical and professional backgrounds].
Based on this information, please answer the following question:

**(a)** As of today, will you vote for the Democratic Party (Hilary Clinton), the Republican Party (Donald Trump), or do you have no preference?

- Democratic
- Republican
- No Preference

---

Here, LLMs were first prompted to generate political ideology features based on demographic inputs. These generated features were then combined with other persona details and used as inputs for the final voting simulation. This two-pipeline design allows us to evaluate the capability of these LLMs in generating relevant features within a political science context. Additionally, it enables us to analyze how these generated features might influence the bias in LLM voting simulations. The following popular general-purpose LLMs are selected as the benchmark models for our experiments: (I) gpt4o-mini-base, (II) gpt4o-base, (III) llama3.1-8B-gen, (IV) gpt4o-mini-gen, (V)gpt4o-gen, (VI) llama3.1-70B-base, (VII) gpt4o-NP, (VIII) llama3.1-8B-base, (IX) llama3.1-70B-gen.

Two evaluation metrics are designed for our empirical study: (1) For voting simulation, we compute the ratio $Ratio = R/(R + D)$, where $R$ and $D$ represent Republican Votes and Democratic Votes, respectively. Then, we compare LLM-generated results with the 2016 ANES dataset Studies (2019). (2) For feature generation, we assess alignment by comparing LLM-generated political ideology features against original ANES features, mapping values from 1 ("Very Liberal") to 7 ("Very Conservative").

### 3.3 Explanation of Evaluation Criteria.

We design two different evaluation criteria to evaluate the bias in voting simulation and the quality of feature generation. For voting simulation, we calculate the ratio: $\mathcal{R} = \frac{\text{Republican Votes}}{\text{Republican Votes}+\text{Democratic Votes}}$ and compare

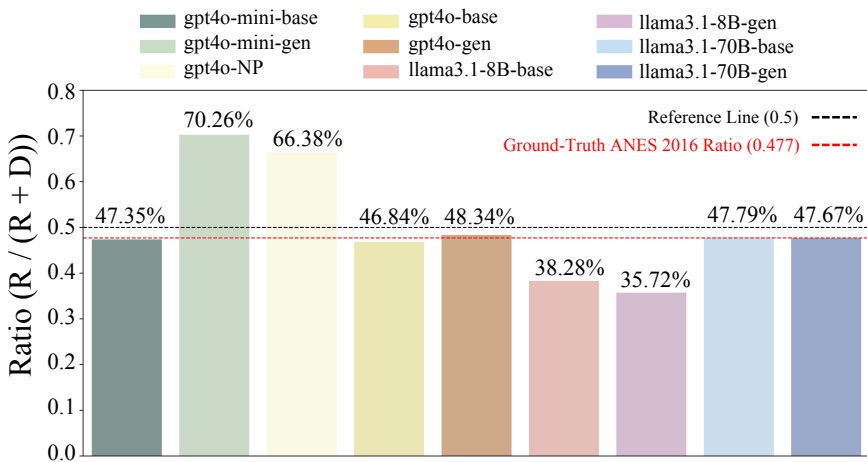

**Figure 3:** Voting simulation results among baseline LLMs on the ANES 2016 benchmark. The chart presents each LLM's predicted voting ratio, showing varying levels of deviation from the ground-truth ratio.

the LLM-generated simulation results with actual outcomes from the 2016 American National Election Studies (ANES) dataset Studies (2019); Dinkelberg et al. (2021).

### 3.4 Results Analysis and Performance Comparison

**Results Analysis from the PPS Perspective.** Figure 3 and Figure 4 provide comprehensive illustrations of the experimental performance comparison among the selected LLMs. Larger models, such as GPT-4o and Llama 3.1-70B, achieve predicted voting ratios that are closely aligned with the ANES reference baseline (47.7%), whereas smaller models exhibit larger deviations. Notably, GPT-4o without using the generated political ideology features skews toward the winning party of the 2016 presidential election, highlighting the need for domain-specific information input to mitigate bias. Figure 4 further demonstrates disparities in the quality of feature generation, with larger-size LLMs producing ideology distributions that more accurately reflect the original dataset, indicated by denser diagonal clusters in (a). In contrast, GPT-4o-mini and Llama 3.1-8B display ideological skewness, favoring dominant political narratives. LLM answer rate analysis in Figure 4 (c) reinforces this trend, with larger-size LLMs achieving response rates exceeding 99%, while smaller-size LLMs show response gaps of up to 7.4%. The results highlight the strengths and limitations of different LLMs in real-world political tasks and show the need for adaptations with larger LLMs.

**Results Analysis from the NPS Perspective.** Beyond analyzing from the PPS perspective that mainly focuses on quantitative measures, the findings below from the NPS perspective offer further insights into applying LLMs in political science. The alignment of larger-size LLMs' prediction with ground-truth data distribution suggests that model scale contributes to fairness and representational balance. Figure 4(a) reveals that ideological distortions in smaller-size LLMs raise concerns about ethical risks when deployed in politically sensitive scenarios. By presenting the response rates of different baseline LLMs, Figure 4(c) highlights the need for more inclusive LLM training schemes, as missing responses often neglect underrepresented viewpoints. These observations underscore the key considerations in normative political science. Furthermore, the varying ideological tendencies of different LLMs suggest that future deployments must incorporate safeguard mechanisms against systemic bias amplification, ensuring responsible and equitable political decision-making.

## 4 Detailed Discussion on Research Outlooks & Open Challenges

### 4.1 A Perspective of Positive Political Science

**(a) Scaling Political Tasks Across Multilingual and Multimodal Contexts.** Political-LLM underscores the challenge of adapting LLMs to diverse linguistic and multimodal political datasets, as explored in Section D.1. Current models often struggle to generalize across languages, cultural nuances, and modalities such as text, video, and audio Pawar et al. (2024). This limitation restricts their ability to provide inclusive

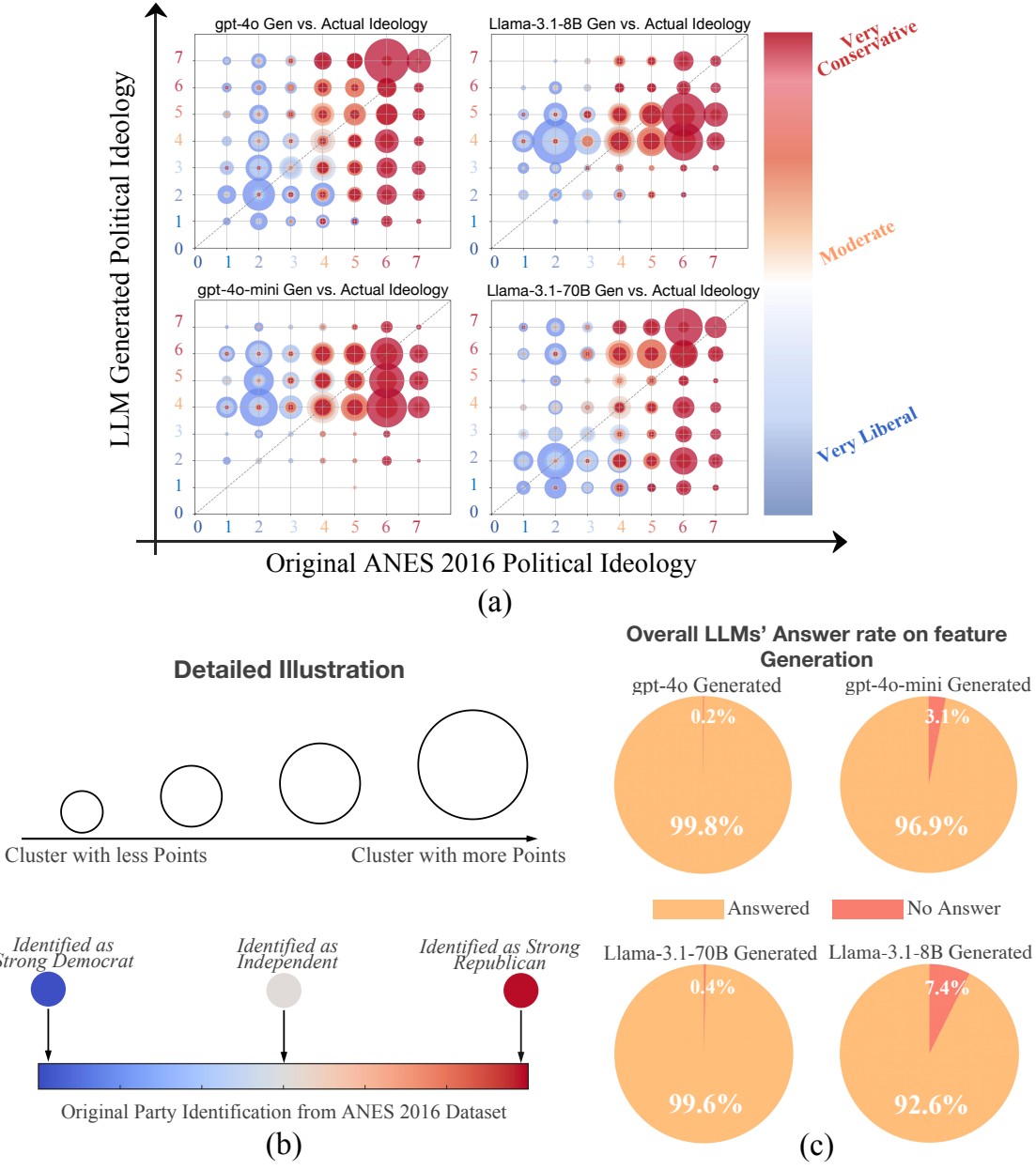

(a)

(b)

(c)

**Figure 4:** Comparison of LLMs in political ideology generation and biases identification. It includes ideology comparison matrices (a), the associated ideological legend for further explanation (b), and the pie charts showing the effectiveness of generation measured with their answering rate (c).

and representative political analysis. To address these challenges, future research must prioritize developing cross-lingual model optimization strategies, such as leveraging multilingual corpora and fine-tuning techniques tailored to political tasks. Moreover, integrating multimodal political data streams, which encompass diverse inputs like legislative texts, speeches, and visual propaganda, requires advanced architectures capable of seamless modality fusion.

**(b) Enhancing LLMs for Policy and Behavioral Simulations.** LLMs hold immense potential for synthesizing policy narratives and simulating voter behavior, as highlighted in Appendix C.2. However, challenges persist in ensuring the contextual relevance and ideological neutrality of outputs Segod et al. (2024). Existing models fail to account for nuanced cultural, demographic, and temporal variables, leading

to oversimplified or biased results Abdurahman et al. (2024). Political-LLM emphasizes the importance of domain-specific datasets and advanced fine-tuning methods to address these challenges. Future research should explore debiasing strategies, dynamic context adaptation, and real-time simulation models that can align generated outputs with complicated political realities. Additionally, incorporating multi-agent frameworks and reinforcement learning can enhance the realism and depth of simulations.

**(c) Simulating Complex Political Dynamics with LLM Agents.** Simulating intricate political dynamics, such as coalition-building and conflict resolution, poses significant challenges (see Section C.3). LLMs usually struggle to capture nuanced inter-agent interactions, power asymmetries, and the influence of external factors on decision-making processes Li et al. (2024a). Current models lack the capability to represent realistic political negotiations or adaptive strategies in dynamic environments. Political-LLM underscores the importance of developing reinforcement learning methods and advanced multi-agent frameworks to address these limitations. Future research should explore integrating domain-specific political knowledge, multi-modal inputs, and real-time information to enhance the accuracy and depth of simulations. Embedding explainability mechanisms will enable researchers to trace the logic behind LLM-driven simulations, fostering transparency and reliability.

**(d) Improving Reasoning and Explainability in Political Applications.** Existing LLMs often function as "black-box" models, making it difficult to trace their reasoning processes, which undermines trust and reliability in high-stakes political applications Coan & Surden (2024). Strengthening reasoning ability is essential for improving model transparency, as structured reasoning helps distinguish logical inference from pattern recognition, ensuring models generate sound political analyses. We advocate for integrating tools like stepwise reasoning mechanisms, attention visualization, and structured argumentation models. Future research should develop domain-specific reasoning techniques tailored to political tasks, ensuring models account for complex interdependencies rather than surface-level correlations. Additionally, human-in-the-loop approaches can refine both reasoning and explainability by incorporating expert validation Mosqueira-Rey et al. (2023).

**(e) Robust Evaluation and Domain-Specific Criteria.** Traditional evaluation metrics fail to capture the complexity and contextual nuances of political science tasks Linegar et al. (2023). Political-LLM highlights the importance of developing proprietary evaluation frameworks that are adaptable to the multifaceted demands of political applications. These frameworks should prioritize developing multidimensional scoring systems that evaluate model performance across various dimensions such as fairness, contextual accuracy, and ideological neutrality, ensuring a comprehensive and nuanced assessment tailored to diverse political scenarios, as emphasized in Section D.1. Moreover, future research should incorporate crowdsourced evaluations to gather diverse feedback from users across various backgrounds Xu et al. (2024a), capturing perceptual differences among groups and enhancing the model's credibility. Expanding these benchmarks to include multilingual and multimodal datasets will further enhance their applicability to global political discourse.

### 4.2 A Perspective of Normative Political Science

**(a) Bias Mitigation and Fairness in Political Applications.** Bias mitigation remains a critical challenge, as evidenced by our empirical analysis using the ANES benchmark, which revealed disparities in performance across demographic groups. These imbalances risk perpetuating systemic inequalities and misrepresenting underrepresented communities in politically sensitive contexts. Political-LLM emphasizes fairness-aware fine-tuning strategies, integrating domain-specific knowledge to address biases at both training and inference stages. Additionally, the framework calls for comprehensive bias quantification techniques to assess equity across linguistic, cultural, and demographic dimensions. We suggest future research prioritizing the creation of globally representative datasets that include low-resource languages and diverse political perspectives, ensuring inclusivity in LLM training. Furthermore, universally adaptable fairness metrics must be developed to provide standardized evaluation criteria for ethical AI deployment in political tasks.

**(b) Promoting Inclusivity Through Multilingual and Culturally Adaptive Models.** We emphasize the critical challenge of linguistic and cultural underrepresentation in current models, which limits their inclusivity and fairness in political analysis Chasalow & Levy (2021). Many existing models are disproportionately trained on high-resource languages, neglecting low-resource linguistic and cultural contexts

essential for equitable political research. Addressing the gaps requires the integration of community-specific datasets, low-resource corpus, and culturally nuanced training strategies to ensure diverse representation. Future research should explore advanced multilingual fine-tuning techniques and adaptive algorithms that allow LLMs to dynamically respond to the intricacies of underrepresented cultures and languages. Moreover, incorporating community-specific datasets help capture localized political discourses and sentiments. Such efforts will pave the way for truly inclusive and culturally adaptive LLMs, fostering equitable participation in global political analysis.

**(c) Ensuring Transparency and Accountability in LLM Predictions.** Transparency is essential in political sensitive applications, such as voter behavior analysis and misinformation detection Pamuk (2024). Political-LLM emphasizes the integration of explainability tools, including attribution mapping, preprocessing audits, and explanation frameworks, to enhance traceability and demystify model decision-making processes. These tools enable researchers to link outputs to specific inputs, making it possible to identify biases or errors in predictions. Future research should explore scalable methodologies to implement these tools in real-time applications, particularly in dynamic political scenarios where accountability is crucial. Collaborative efforts between computer science developers and political scientists will further refine these techniques, fostering ethical and informed use of LLMs in decision-making.

**(d) Integrating Ethical Standards for Responsible LLM Deployment.** Ethical concerns, including misinformation risks, ideological biases, and the amplification of harmful narratives, demand the establishment of rigorous guidelines for the development and deployment of LLMs (Appendix C.5). We advocate for cross-disciplinary collaborations involving political scientists, ethicists, and AI researchers to co-create robust evaluation criteria focused on transparency, neutrality, and adherence to democratic principles. Ethical auditing mechanisms, such as periodic assessments of model behavior and proactive safeguards like bias detection algorithms, should be prioritized to ensure alignment with societal values. Additionally, domain-specific ethical benchmarks should be developed for political sensitive tasks, such as election forecasting and policy impact analysis. Future research should explore scalable governance structures and real-time monitoring systems that enhance the responsible use of LLMs and minimize risks in high-stakes political applications.

**(e) Transforming Societal Impacts Through Responsible AI Governance.** Existing governance framework for LLM in political science lack the required adaptability to address the rapid dissemination of false information in diverse political scenarios. Our findings highlight the importance of developing real-time misinformation detection mechanisms, complemented by robust public accountability systems to ensure transparency in high-stakes applications (Appendix C.5). The systems should include audit trails for model outputs and interactive platforms for verifying generated content. We suggest that future research should also focus on integrating compliance supervision into the lifecycle of an LLM, including proactive monitoring and collaborative policymaking involving stakeholders from academia, civil society, and government. By fostering societal trust and enabling democratic engagement, these measures ensure LLMs contribute positively to the political landscape while minimizing risks associated with their misuse.

## 5 Conclusions

This work marks the first comprehensive interdisciplinary study integrating LLMs into political science, bridging traditional methodologies with modern computational techniques. We propose a principled taxonomy categorizing LLM-driven political tasks into Positive and Normative dimensions, offering a structured framework to align the transformative potential of LLMs with the field's unique demands. We highlight LLM applications in predictive modeling, generative tasks, simulation, and causal reasoning, alongside ethical and societal considerations. Our empirical study on ANES benchmark illustrates the practical value of integrating LLMs into political science. To address substantial challenges such as data scarcity, biases, and explainability limitations, we advocate for proprietary benchmarks, tailored training strategies, and robust evaluation metrics. This research underscores the need for interdisciplinary collaboration to navigate both opportunities and challenges, fostering responsible, transparent, and equitable integration of LLMs into political science.

## Limitations

Although Political-LLM provides a principled taxonomy, an in-depth analysis of observations and insights, and empirical validation for integrating Large Language Models into political science, several areas warrant further refinement. First, while comprehensive and systematic, our taxonomy may require additional considerations for emerging political tasks and region-specific implementations. Second, the empirical study primarily focuses on voting simulations, whereas broader validation across diverse political applications, such as legislative analysis and policy deliberation, remains an avenue for future research. Finally, while we advocate for domain-specific evaluation criteria, their full implementation and standardization across interdisciplinary research communities require further efforts. These limitations present opportunities for continued explorations.

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

# A   Preliminaries

**Computational Political Science (CPS).** Computational Political Science (CPS) is an interdisciplinary field that integrates computational methods with political science to analyze political systems, behaviors, and outcomes Haq et al. (2020); Hu et al. (2025). By leveraging tools such as data analytics, machine learning, and natural language processing (NLP), CPS enhances the understanding of complex political phenomena. The field has evolved from relying on traditional statistical models, such as regression-based analyses, to embracing AI-driven approaches that enable the processing of large-scale, unstructured political data. This shift has been particularly transformative in tasks like election forecasting, public opinion analysis, and policy evaluation, where modern techniques offer greater scalability and accuracy.

**Evolution of Language Models in Political Science.** Early applications of AI in political science relied on rule-based systems and traditional machine learning methods Grimmer et al. (2021), such as logistic regression Nicolau (2007) and support vector machines d'Orazio et al. (2014), to perform basic political tasks. These methods were limited by their reliance on manually crafted features and structured data Grimmer et al. (2021). The advent of pre-trained language models, including Word2Vec Mikolov et al. (2013) and BERT Devlin (2018), marked a significant shift in natural language processing, enabling the analysis of large-scale, unstructured political text. By capturing contextual relationships and semantic nuances in data, these models greatly enhanced the ability to process complex political discourse, advancing domain applications like policy analysis, legislative interpretation, and public opinion mining.

## A.1   Large Language Models (LLMs)

The foundation of most LLMs lies in the Transformer architecture Vaswani et al. (2017), which introduced the self-attention mechanism to effectively model long-range dependencies in text. This innovation marked a departure from earlier sequence models like RNNs and LSTMs, which struggled with vanishing gradients and limited context windows. Core components such as multi-head attention, feedforward layers, and positional encodings enabled Transformers to process sequences in parallel, significantly improving scalability and efficiency. Early LLMs, such as BERT Devlin (2018), leveraged the Transformer framework through masked language modeling, excelling in bidirectional context understanding. Autoregressive architectures like GPT Brown et al. (2020) later extended these capabilities, focusing on sequential token prediction for fluent and coherent text generation. The advent of models like T5 Raffel et al. (2020) unified various NLP tasks under a single architecture by using sequence-to-sequence learning. Recent advancements Salemi & Zamani (2024); Kirk et al. (2024); Song et al. (2024a); Zhu et al. (2024) further evolved LLM architectures, emphasizing efficiency and task-specific adaptability. Additionally, innovations like multimodal architectures and scalable models such as LLaMA Touvron et al. (2023) and GPT-4 Achiam et al. (2023) demonstrate a shift toward systems capable of cross-domain understanding and dynamic interaction, underpinning the transformative potential of LLMs in computational tasks across fields.

**General-Purpose LLMs.** General-purpose LLMs like GPT Achiam et al. (2023) and BERT Devlin (2018) are developed through two primary training paradigms: autoregressive modeling (AR) Schuurmans et al. (2024) and masked language modeling (MLM) Nozza et al. (2020). AR models, exemplified by GPT, generate tokens sequentially, prioritizing fluency and coherence in text generation. MLM, as utilized in BERT, predicts masked tokens within sentences, fostering a nuanced contextual understanding of language. These pre-training paradigms equip LLMs with a robust foundational understanding of linguistic patterns, making them highly adaptable for fine-tuning on task-specific datasets. Leveraging these versatile models, researchers can efficiently address domain-specific challenges without undertaking resource-intensive pre-training.

**LLM Fine-tuning Techniques.** Fine-tuning adapts pre-trained general-purpose LLMs to downstream specialized applications. Supervised fine-tuning refines model outputs using labeled datasets, aligning them with task-specific objectives Li et al. (2023). Instruction fine-tuning trains models to better follow user directives through datasets of instructions and outputs, enhancing adherence to user intents and versatility Zhang et al. (2023b). Reinforcement Learning with Human Feedback (RLHF) Lee et al. (2024a) leverages human evaluators to rank responses, guiding LLMs to align with human preferences while reducing harmful or biased behaviors.

**Zero-shot, Few-shot, and In-context Learning.** LLMs demonstrate remarkable capabilities in zero-shot Kojima et al. (2022), few-shot Perez et al. (2021), and in-context learning Ram et al. (2023), leveraging pre-trained knowledge to perform tasks with minimal or no additional training. Zero-shot learning enables task generalization without task-specific training, while few-shot learning benefits from a minimal set of labeled examples. In-context learning, which is achieved through task descriptions and examples within prompts, empowers models to dynamically adapt to novel tasks without parameter updates.

**LLM Inference and Decoding Techniques.** Effective inference strategies are crucial for generating high-quality outputs from LLMs. Methods like greedy decoding Prabhu (2024) and beam search Xie et al. (2024b) prioritize sequence coherence, while nucleus sampling Grubisic et al. (2024) enhances diversity by sampling within the top-probability distribution. Advanced techniques like Retrieval-Augmented Generation (RAG) Salemi & Zamani (2024) integrate external knowledge bases, while prompt engineering White et al. (2023), Chain-of-Thought (CoT) Yao et al. (2024b), and knowledge injection techniques Martino et al. (2023) improve task-specific performance, especially in complex scenarios.

**LLM Scalability and Efficiency.** LLM scalability relies on distributed training frameworks Narayanan et al. (2021), efficient parameter adaptation Ding et al. (2023), fast inference and serving frameworks Wu et al. (2023a), and hardware optimizations Song et al. (2024b). Techniques like LoRA Ren et al. (2024) and adapters Fu et al. (2023) enable parameter-efficient fine-tuning, reducing computational requirements without compromising performance. Software frameworks such as vLLM Kwon et al. (2023) and TensorRT-LLM NVIDIA (2024) facilitate fast LLM inference and serving through advanced batching and memory management. Hardware acceleration, including GPU and TPU advancements Song et al. (2024b); Wu et al. (2023a), supports the training and inference of increasingly large models, driving efficiency in both computation and energy consumption.

## A.2 Core Computational Political Science Concepts

**Political Data Sources and Text Generation.** Political data encompasses diverse sources such as political news, speeches, legislative records, party manifestos, social media content, etc. Analyzing these data requires handling challenges like data scarcity, imbalance, and linguistic nuances, which hinder comprehensive analysis. One critical application in CPS is *Political Text Generation*, where LLMs are employed to produce political content such as speeches, policy briefs, and debate scripts Zhang et al. (2023a); Churina & Jaidka (2024). These generative models assist political figures and analysts by creating coherent, persuasive, and contextually relevant texts. LLMs can simulate political scenarios and craft narratives, shaping public opinion and enhancing political communication.

**Election Prediction and Voting Behavior.** Election prediction focuses on forecasting voter turnout, swing state dynamics, and overall electoral outcomes. LLMs analyze a combination of historical election data, public opinion surveys, and social media discourse to identify patterns influencing voter behavior Rotaru et al. (2024); Potter et al. (2024). These models provide insights into key demographic and psychological factors affecting voter preferences, aiding political campaigns and policymakers in tailoring strategies to engage the electorate effectively.

**Policy and Legislative Interpretation.** Policy and legislative interpretation involve analyzing complex legal texts, such as bills, statutes, administrative rules, and debates, to understand their implications and the ideologies they represent Jiao et al. (2025). LLMs excel at parsing and summarizing these documents, identifying key arguments, and predicting potential policy outcomes Cheong et al. (2024). This capability offers political scientists a deeper understanding of legislative processes and helps anticipate the effects of policy changes on societal and political structures.

**Misinformation/Fake News Detection.** Safeguarding political discourse requires addressing misinformation and fake news, which can significantly distort public opinion and decision-making Baron & Ish-Shalom (2024). LLMs are adept at detecting false or biased information by analyzing the structure, intent, and credibility of news articles, social media posts, and political statements Wu et al. (2024a). By flagging harmful content, these models ensure the integrity of political information and contribute to maintaining a healthy democratic environment.

**Political Risk and Conflict Prediction.** Political risk and conflict prediction aim to forecast the likelihood of political instability, unrest, or international conflict Croicu & von der Maase (2025). CPS-based methods can analyze geopolitical data, diplomatic communications, and historical trends to identify early signs of conflict and assess the risks involved in political decisions. These predictions are invaluable for policymakers and international organizations in making informed decisions and preparing for possible crises.

**Political Game Theory and Negotiation.** Political game theory and negotiation involve modeling strategic interactions between political entities, such as governments, parties, or international entities Fundamental AI Research Diplomacy Team et al. (2022); Gaffal & Padilla Gálvez (2024). The latest advancements in LLMs hold promise in analyzing negotiation strategies, predicting the outcomes of political bargaining, and identifying optimal decision-making approaches. By simulating various political scenarios, LLMs are expected to have a better understanding of power dynamics, coalition building, and diplomatic negotiations in international politics.

LLMs act as pivotal tools bridging computational methodologies and political science applications. By integrating advanced language processing capabilities with political data analysis, LLMs enable breakthroughs in tasks such as election forecasting Rotaru et al. (2024), legislative text summarization Colombo (2024), and combating misinformation Wu et al. (2024a). These advancements demonstrate how LLMs transcend traditional limitations, providing scalable, adaptable, and effective solutions for political science research.

# B Detailed Comparison between Political-LLM and Existing Studies on Broad Political Science Field

**Table 1:** Comparison with Existing Studies in Broad Political Science Field (Abbreviations: PoliSci = Political Science, CPS = Computational Political Science, CS = Computer Science).

| | Ziems et al. (2024) | Argyle et al. (2023b) | Ornstein et al. (2022) | Rozado (2023) | Weidinger et al. (2021) | Linegar et al. (2023) | Lee et al. (2024b) | Political-LLM (Ours) |
|---|---|---|---|---|---|---|---|---|
| Proposed Taxonomy on LLM for PoliSci | ✗ | ✗ | ✗ | ✗ | ✗ | ✗ | ✗ | ✓ |
| Literature Review from PoliSci Perspective | ✓ | ✗ | ✓ | ✗ | ✗ | ✓ | ✓ | ✓ |
| Literature Review from CS Perspective | ✗ | ✗ | ✓ | ✗ | ✓ | ✓ | ✗ | ✓ |
| Structured Analysis of CPS Methodologies | ✗ | ✗ | ✗ | ✗ | ✗ | ✗ | ✗ | ✓ |
| Include Experiments and Evaluations | ✓ | ✓ | ✓ | ✓ | ✗ | ✗ | ✓ | ✓ |
| Application Examples | ✓ | ✓ | ✓ | ✓ | ✓ | ✓ | ✓ | ✓ |
| Comprehensive Summary of Benchmarks | ✓ | ✗ | ✗ | ✗ | ✗ | ✗ | ✗ | ✓ |
| Analyzing Limitations in Existing Methodologies | ✓ | ✓ | ✓ | ✓ | ✓ | ✓ | ✓ | ✓ |
| Future Research Direction | ✓ | ✓ | ✓ | ✓ | ✓ | ✓ | ✗ | ✓ |

Despite the abundant explorations in this interdisciplinary area, current survey works remain limited. Researchers have realized and discussed the potential revolutionary contribution of language models as early as in 2017 Wilkerson & Casas (2017); Terechshenko et al. (2020). However, these works mainly focus on traditional language models and are unable to provide insights about more recent LLMs. After that, multiple survey works have realized the potential of LLMs for political science Chatsiou & Mikhaylov (2020); Rodman (2024). However, their discussion lacks a systematic understanding of how LLMs can be adopted in various political science applications and research. More recently, LLM-based applications on specific political or social tasks have been reviewed in several survey works Lee et al. (2024b); Argyle et al. (2023b); Linegar et al. (2023); Rozado (2023); Ziems et al. (2024); Ornstein et al. (2022); Weidinger et al. (2021). Nevertheless,

these works overwhelmingly focus on applications while the discussion from a technical perspective is ignored. Therefore, it remains unclear how LLMs can be improved to be better adapted. Different from all the survey works above, we aim to present a systematic and comprehensive understanding of leveraging LLM's power for political science. Specifically, we equip this paper with a novel principled taxonomy to classify existing works, such that researchers and practitioners can have a broader picture of this interdisciplinary field. Meanwhile, we perform a discussion on each type of work from both political and technical perspectives, which reveals how LLMs can be improved to be better adapted. Table 1 provides a detailed comparison between our survey and other related surveys in political or social science.

## C    Classical Political Science Functions and Modern Transformations

LLMs have brought transformative changes to political science, reshaping traditional methodologies and unlocking new analytical opportunities. This section provides a structured overview of current research, categorizing it into five key areas. Four of these areas focus on the functional applications of LLMs in political science, while the fifth explores normative considerations, emphasizing societal and ethical implications.

We divide the functional categories into prediction, generation, simulation, and explainability tasks. While computer science researchers often categorize LLM-based research into predictive and generative tasks Demszky et al. (2023); Khurana et al. (2023); Minaee et al. (2024), we propose two additional dimensions - simulation and explainability, in order to address the unique complexity of LLM for Political Science.

While simulation is inherently generative, we distinguish it as a separate category due to its unique focus on replicating human-like attitudes, behaviors, and decision-making processes in specific political scenarios. In this review, we list research that focuses on producing new content without emulating human cognitive processes as "generative tasks", and research mimicking how human actors or groups would react, taking into account motivations, biases, and contextual influences as "simulation".

Additionally, political science is not merely concerned with making predictions but also with understanding the causes behind political phenomena. For instance, in addition to predicting the outcome of elections, political scientists are also interested in why certain outcomes occur. Therefore, using LLMs to support inference tasks (e.g., processing vast datasets, and identifying causal mechanisms) is promising in political science and a necessary addition to predictive and generative tasks.

### C.1    Automation of Predictive Tasks

**Definition.** Predictive tasks in Computational Political Science involve anticipating future events or trends based on existing data, and they are fundamental in political science for applications such as election forecasting, policy support prediction, and analyzing voter behavior. In political science, predictive tasks are crucial because they provide insights that can guide decision-making, inform policy, and help researchers understand complex social dynamics. Traditional predictive methods in political science often require extensive manual labor. For instance, certain predictive tasks may require researchers to manually collect survey responses, historical election data, or economic indicators, which can be time-consuming and prone to human error. In contrast, recent advancements in LLMs offer an alternative by automating predictive tasks. The automation of predictive tasks reduces manual effort and possible human error, while increasing speed, consistency, and scalability.

**Enhancing Prediction with LLM-based Data Annotation.** LLM-based automation significantly enhances predictive capabilities by providing consistent and scalable solutions for data-intensive tasks. This is especially helpful in data annotation. Annotating large datasets manually is time-consuming and prone to inconsistencies Heseltine & Clemm von Hohenberg (2024); Liu & Shi (2024); Egami et al. (2024). LLMs can rapidly process and annotate data in a consistent manner. Political science researchers have employed LLMs to annotate *Political Ideology* Heseltine & Clemm von Hohenberg (2024); Liu et al. (2022); Chalkidis & Brandl (2024); Cao et al. (2024), *Fake News* Wang et al. (2024a); Wu et al. (2024b); Hu et al. (2024); Whitehouse et al. (2022), *Tone (sentiment)* Heseltine & Clemm von Hohenberg (2024); Liu & Shi (2024); Cao et al. (2024); Fu et al. (2024); Lashitew & Mu (2024), and content of various *Political Texts* Heseltine & Clemm von Hohenberg (2024); Liu & Shi (2024); Kocielnik et al. (2023); Gambini et al. (2024); Cao et al.

(2024). Researchers also find that the quality of automated LLM annotation outperforms crowd workers and even some domain experts. Therefore, data annotation by LLM not only enhances efficiency but also reduces the potential for human bias and error in the data annotation process.

**Prediction Tasks in English-Speaking Contexts.** The effectiveness of LLMs in predictive tasks is demonstrated through their applications in both English and non-English contexts. In English-speaking settings, platforms like ChatGPT and Llama are frequently used for large-scale political text analysis. For instance, Lashitew and Mu Lashitew & Mu (2024) analyze comments and letters submitted by companies to the U.S. Securities and Exchange Commission regarding climate change disclosure regulations. Leveraging GPT-3, they efficiently process and analyze a large volume of text data, identifying patterns and sentiments within the corporate responses. Additionally, Fu et al. (2024) explore the application of GPT-4 in processing and analyzing public feedback collected online in New Zealand. They focus on responses to a proposed plan change in Hamilton City, New Zealand, assessing GPT-4's effectiveness in summarizing feedback, identifying topics, and analyzing sentiment. Results showed GPT-4 performed these tasks accurately.

**Predictive Tasks in Non-English Contexts.** LLMs have also shown robust performance in multilingual environments and in diverse regional applications. Heseltine & Clemm von Hohenberg (2024) evaluate the performance of GPT-4 in coding political texts across variables such as relevance, negativity, sentiment, and ideology across the United States, Chile, Germany, and Italy. The findings indicate that GPT-4's annotations closely align with those of human experts, suggesting that LLMs can effectively assist in political text analysis. Moreover, Chalkidis and Brandl Chalkidis & Brandl (2024) utilize Llama to evaluate speeches from European Parliament debates, with the EUandI questionnaire serving as a reference or benchmark to verify political leanings. The study demonstrated that Llama has considerable knowledge of national parties' positions and is capable of contextual reasoning as well as ChatGPT. Mellon et al. Mellon et al. (2024) take a step further to evaluate six different popular LLMs in categorizing open-text survey responses and detecting issue importance. Their task involved classifying the most important issue responses from the British Election Study Internet Panel into 50 distinct categories. The study concluded that LLMs, particularly Claude-1.3, can effectively code open-text survey responses, providing a scalable alternative to human coders.

**LLM-based Advancements.** To better illustrate the workflow of LLMs in predictive tasks, we provide a diagram showcasing the U.S. Presidential Election outcome prediction as an example. This example highlights how LLMs integrate diverse data sources, process them into structured representations, and generate actionable predictions. As shown in Figure 5, the workflow begins by integrating data sources like polls, demographics, social media sentiment, and news headlines. After preprocessing (cleaning, normalization, and vectorization), the LLM performs contextual understanding and generates outputs such as winning probabilities and swing state predictions, showcasing its ability to automate complex, data-driven tasks like U.S. election predictions.

In addition to predictive applications in various domains, recent research also highlights tailored frameworks and approaches developed specifically for political science. Such research often includes adjustments to LLMs to improve their applicability and accuracy in political scenarios. For instance, PoliPrompt Liu & Shi (2024) is a three-stage framework leveraging LLMs for text classification in political science. This framework shows exceptional performance in classifying topics within multi-class news datasets, such as BBC news reports, labeling nuanced political science concepts, and analyzing the tones of campaign advertisements from the 2018 midterm election. This kind of tailored approach helps ensure that the model outputs are relevant and accurate within specific political frameworks. Similarly, by studying the classification on text alignment or opposition toward a particular issue, Cao and Drinkle Cao et al. (2024) find that incorporating metadata (e.g., party affiliation) into political stance detection tasks can notably enhance model performance on ParlVote+ benchmark.

**Summary and Challenges.** The automation of predictive tasks by LLMs offers transformative potential in political science research Lazar & Manuali (2024); Linegar et al. (2023). From scaling data annotation processes to handling multilingual data and adapting to specific political frameworks, LLMs provide a powerful tool for researchers aiming to predict and analyze trends in political behavior and sentiment, as well as test political theories. However, existing research still faces notable challenges. LLMs can sometimes lack contextual understanding in nuanced political discourse, particularly in multilingual or culturally specific

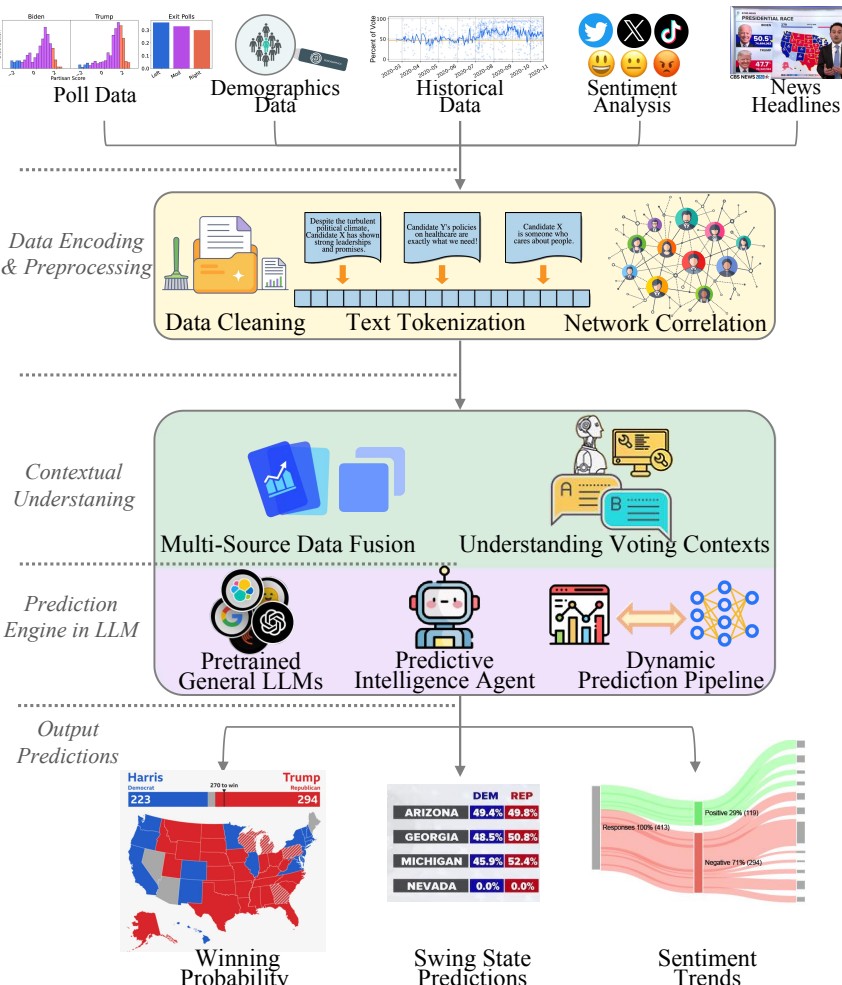

**Figure 5:** The workflow of LLM-based automated predictive task, using the U.S. Presidential Election prediction as an example.

settings, where subtle language differences may lead to misinterpretation He et al. (2023). Additionally, the reliance on pre-existing data and the potential for inherent biases in training datasets can result in biased predictions, impacting the accuracy and neutrality of the automated outputs Bang et al. (2024). Furthermore, while LLMs have shown proficiency in annotation and classification, their performance may degrade when faced with highly complex or specialized political tasks that require deeper domain knowledge Wu et al. (2023b). Addressing these limitations is essential for maximizing the utility and reliability of LLMs in political science applications.

## C.2 Automation of Generative Tasks

**Definition.** Generative tasks in political science involve creating synthetic data, simulating scenarios, or augmenting incomplete datasets, offering new insights where traditional data sources are either unavailable or insufficient Argyle et al. (2023b); Bisbee et al. (2024); Wu et al. (2023b); Napolio (2024). Unlike analytical tasks that focus on interpreting existing information, generative tasks expand the boundaries of what can be studied by creating representations of missing data or by projecting possible future scenarios Argyle et al. (2023b); Bisbee et al. (2024); Wu et al. (2023b). Generative tasks are particularly valuable for political science applications where the complete dataset is hard to obtain due to privacy concerns, logistical constraints, or high costs associated with traditional data collection methodologies Napolio (2024); Palmer & Spirling (2023).

The absence of complete data often underscores the complexity, scope, and depth of political science research questions Argyle et al. (2023b); Bisbee et al. (2024); Wu et al. (2023b); Napolio (2024). For example, understanding the roles and performance of executive agencies, which exert significant influence over policy, presents substantial challenges due to the limited availability of data Napolio (2024). Traditional CPS approaches, such as principal component analysis (PCA)-based methods, demand extensive input data, limiting their application in issue-specific analyses, such as polarizing topics like abortion or gun control. LLMs are capable of extracting valuable insights from incomplete datasets if provided with well-structured prompts, broadening the analytical capacity of studies Argyle et al. (2023b); Bisbee et al. (2024); Wu et al. (2023b). This innovation enables the exploration of previously constrained research areas Napolio (2024); Palmer & Spirling (2023). Existing research in this domain can be grouped into two major categories: *Synthesizing Political Data* and *Enhancing Research Scope.*

**Synthesizing Political Data.** The ability to generate synthetic data is a powerful application that directly addresses the critical issue of data scarcity and facilitates the exploration of latent variables. Data collection is often a significant hurdle in political science due to the costs and time involved in conducting surveys, gathering reliable public opinion data, or accessing confidential voting records. Synthetic data generation by LLMs offers an efficient, cost-effective alternative that can serve as a proxy for real-world data, providing insights where traditional data sources are limited Wang et al. (2024c). For instance, Bisbee et al. Bisbee et al. (2024) demonstrate that LLM-generated synthetic data can effectively replicate survey responses, simulating various public opinion trends even in the absence of comprehensive survey datasets. They successfully explore public sentiment on immigration, healthcare, and climate policy issues. This application is particularly useful for analyzing time-sensitive political questions, where delays in data collection could mean losing valuable insights into changing public opinion. Another noteworthy study comes from Argyle et al. Argyle et al. (2023b), who show that LLMs can simulate human responses, mimicking the distribution of survey data across demographic groups and regions. In this case, LLMs help mitigate the data scarcity issue by generating synthetic samples that reflect genuine population characteristics, supporting research on political trends in underserved or underrepresented communities. We provide the workflow of LLM-based generative tasks in Figure 6, using the synthesis of political speeches or manifestos as an example. Starting with inputs like topic definitions, ideological tags, and tone preferences, the model preprocesses and contextualizes data to generate coherent outputs. Techniques such as prompt engineering and fine-tuning guide the process. The outputs, including political speeches tailored to ideological perspectives, demonstrate how LLMs can address challenges of data scarcity and enable synthetic data generation for political research.

LLMs also play a critical role in estimating political ideologies in situations where conventional data sources, such as voting records, media publications, or public statements, are incomplete. Wu et al. Wu et al. (2023b) illustrate how LLMs can infer political ideologies by analyzing existing contextual information and filling in missing details, thereby offering a fuller, more nuanced picture of the ideological spectrum in specific political landscapes. Moreover, Alvarez et al. Alvarez et al. (2023) explore the potential of LLMs in simulating voter behavior and party strategies, thus extending traditional political modeling frameworks. By generating synthetic data that represents hypothetical voter responses to specific policies or campaign strategies, LLMs help researchers examine potential outcomes in elections or other political events. Such applications offer new avenues for understanding the impact of political campaigns and policy proposals, even when comprehensive polling data is unavailable.

LLMs further enhance research potential by enabling the generation of large and dynamic datasets that track the latest political trends over time. Palmer & Spirling (2023) emphasize the utility of LLMs in constructing extensive synthetic datasets by generating responses or synthesizing textual data. This enables the analysis of long-term shifts in public opinion or political rhetoric across diverse populations.

**Enhancing Research Scope.** Beyond data synthesis, LLMs enable researchers to explore previously unattainable research areas by providing insights into complex or hard-to-measure variables. This capacity to expand the scope of political science research is especially valuable in analyzing intricate social dynamics, government policies, and ideological nuances where data gaps often hinder rigorous analysis. For example, Napolio's Napolio (2024) work on the ideological positioning of executive agencies illustrates how LLMs can provide insights into policy stances and organizational biases even in the absence of direct, comprehensive data. The use of LLMs to fill data gaps allows for a deeper understanding of government operations and

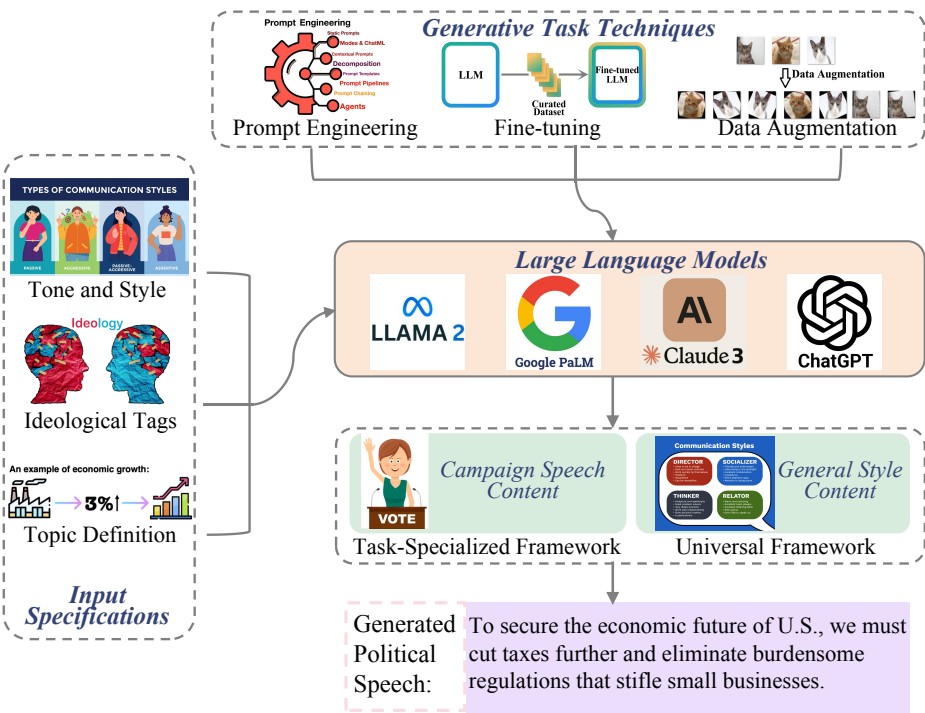

**Figure 6:** Workflow for LLM-based generative tasks, illustrating the synthesis of political speeches with specific ideology, style, and focus of content.

policy influences that would otherwise remain hidden. Similarly, Egami et al. Egami et al. (2024) demonstrate that LLMs can work with imperfect or noisy data, producing robust analytical results even when complete datasets are unavailable. This flexibility reduces dependency on high-quality data and supports rigorous analysis in fields like public policy and election studies, where data completeness is challenging to achieve.

LLMs are also adept at analyzing extensive political text corpora, which enables researchers to uncover subtle patterns in discourse that are difficult to capture through traditional manual analysis. Palmer and Spirling Palmer & Spirling (2023) highlights the ability of LLMs to process large volumes of text, revealing shifts in political narratives and public sentiment over time. Similarly, the use of LLMs to analyze political Q&A sessions in Alvarez & Morrier (2024) shows how these models can detect nuances in rhetoric and speaker intent, providing valuable insights into the subtleties of political communication. Furthermore, Mellon et al. Mellon et al. (2024) showcase the utility of LLMs in coding open-ended survey responses at scale. This application allows researchers to classify responses efficiently, identifying dominant issues and sentiments within a population. By automating the analysis of qualitative data, LLMs offer a powerful solution for understanding public concerns and policy impacts, contributing to a more comprehensive understanding of societal dynamics.

**Summary and Challenges.** LLMs have reshaped the field of generative tasks in political science, enabling new possibilities in data synthesis and research scopes. These models provide political scientists with the tools needed to address data scarcity issues, create realistic proxies for hard-to-collect data, and simulate complex political phenomena. However, the challenges in ensuring the validity, neutrality, and reliability of synthetic data remain significant. Biases embedded in LLM-generated data can potentially skew results if not rigorously managed, and reliance on synthetic data requires careful validation to ensure accuracy. Moreover, while LLMs are proficient in generating insights, the explainability of these models in highly nuanced contexts of political science remains a challenge. Addressing these limitations will be essential for leveraging the full potential of LLMs in generative political science research.

### C.3 Simulation of LLM Agents

**Definitions.** The concept of Simulation Agents in LLM for political science refers to the use of large language models to create interactive environments in which autonomous agents simulate behaviors, decisions, or dialogues. These tasks aim to explore dynamic systems, such as political behaviors, negotiations, or conflicts, by modeling interactions between agents. While both Generative Tasks and Simulation Agents leverage LLMs, their objectives and methodologies are distinct Mou et al. (2024a). Generative tasks focus on creating new data or textual content to address data scarcity, enabling researchers to fill gaps or produce synthetic datasets for foundational analysis. In contrast, simulation agents emphasize modeling interactions and dynamics within complex environments, offering insights into strategies, behaviors, and evolving systems. We present a comprehensive comparison between these tasks in Table 2.

**Table 2:** Comparison of Generative Tasks and Simulation Tasks in Political Science

| Key Attribute | Generative Tasks | Simulation Tasks |
|---|---|---|
| *Objective* | Create new data or textual content to address data scarcity. | Model interactions and dynamics within complex environments. |
| *Focus* | Producing synthetic data or content for foundational analysis. | Exploring strategies, behaviors, or evolving systems through agents. |
| *Output* | Independent generated results, such as datasets or textual outputs. | Analytical results on interactions, strategies, or behavior patterns. |
| *Research Context* | Filling data gaps and enhancing data availability. | Studying dynamic processes and agent interactions. |
| *Methodology* | Generative models producing outputs based on prompts. | Simulations of agents interacting within predefined environments. |
| *Application Examples* | Synthetic survey data, opinion generation, or text classification. | Negotiation models, conflict dynamics, or opinion shift simulations. |

The use of LLMs to simulate human-like behavior in interactive environments represents a significant advancement in political science Park et al. (2023). These simulations offer new ways to address complex societal questions, particularly those involving the behavior of political actors in intricate environments De Marchi & Page (2014). Traditional methods, such as Agent-Based Models (ABMs) De Marchi (2005), rely on predefined parameters and restricted environments, often limiting their capacity to capture the complexity and realism of political dynamics. LLMs overcome these constraints by using natural language prompts to define behavior rules and environmental contexts Gao et al. (2023), allowing for adaptive, context-sensitive, and personalized agent behaviors Wang et al. (2024b). Current research in this area is focused primarily on two applications: (1) *using agents to simulate behavior dynamics* and (2) *using agents to simulate text-based discussion processes* Guan et al. (2024); Moghimifar et al. (2024); Wang et al. (2024b).

**Simulate Behavior Dynamics.** Recent studies demonstrate the potential of LLMs to replicate complex social behaviors in political settings, addressing limitations in traditional ABM approaches. Dai et al. Dai et al. (2024) simulate agents shifting from conflict to cooperation in resource-constrained environments through Hobbesian Social Contract Theory, exploring how political entities navigate scarcity and develop governance structures. Hua et al. Hua et al. (2023) take a historical approach, modeling strategic decision-making during major global conflicts such as the World Wars, focusing on the interplay between diplomacy and military tactics in the evolution of warfare. Jin et al. Jin et al. (2024) extend these simulations to a cosmic scale, where agents with distinct worldviews engage in cooperation and conflict, highlighting how ideological divergence influences inter-civilization dynamics. Other research builds on these approaches by introducing more nuanced political scenarios. Chuang et al. Chuang et al. (2023) simulate opinion dynamics within political networks, where agents adjust their beliefs based on interactions with other agents, providing a closer examination of political polarization and consensus-building processes. Similarly, Guan et al. Guan et al. (2024) use LLM-based agents to model AI diplomacy, where agents negotiate and evolve their strategies in complex international relations, mirroring real-world diplomatic negotiations. These studies collectively showcase how LLM-driven simulations of behavior dynamics can provide valuable insights into governance, conflict resolution, and social interaction, offering novel ways to study political and diplomatic behavior in

various contexts Dai et al. (2024); Guan et al. (2024); Yao et al. (2024a); Jin et al. (2024); Chuang et al. (2023); Hua et al. (2023).

**Simulate Text-based Discussion.** Shifting from physical to text-based simulations, recent studies have explored political interactions through dialogue, using LLM agents to simulate complex discussions and negotiations. Baker et al. Baker & Azher (2024) model U.S. Senate policy debates, where LLM agents simulate legislative decision-making and bipartisanship, providing insights into how political actors navigate ideological divides and negotiate policy outcomes. Moghimifar et al. Moghimifar et al. (2024) take a different approach by simulating multi-party coalition negotiations using LLM-driven dialogue. Their work highlights the intricacies of building and maintaining political alliances through textual interaction, emphasizing how agents negotiate, compromise, and form agreements in multi-party systems. Guan et al. Guan et al. (2024) extend this approach to international diplomacy, focusing on how LLM agents evolve strategies in alliance-building and negotiation on the global stage. Their research underscores the dynamic nature of diplomatic discourse, where agents adapt to shifting geopolitical scenarios and evolving relationships between states. Additionally, Jin et al. Jin et al. (2024) explore the use of LLMs in simulating text-based discussions between civilizations with divergent worldviews, pushing the boundaries of how text-based interactions can simulate inter-group communication and conflict resolution on a cosmic scale. These studies collectively illustrate the capacity of LLM simulations to model political decision-making through textual interactions, offering a contrast to action-oriented simulations like those seen in warfare and conflict resolution Baker & Azher (2024); Chuang et al. (2023); Moghimifar et al. (2024); Guan et al. (2024); Jin et al. (2024).

**Summary and Challenges.** LLM-driven simulations provide a novel framework for exploring the complexity of political behavior and interactions by enabling adaptive, context-aware modeling that was previously unattainable with traditional methods. These simulations bridge gaps in understanding how dynamic processes, such as opinion shifts, negotiation strategies, and conflict resolution, evolve under different political scenarios. Despite these advancements, significant challenges persist. Ensuring the neutrality of simulations remains difficult due to biases inherent in LLM training datasets, which can skew outcomes and interpretations. Moreover, ethical concerns arise when simulations replicate sensitive behaviors or policy decisions, potentially influencing real-world political discourse. Last but not least, the computational costs of running large-scale simulations can limit accessibility for many researchers. Addressing these issues will require robust validation techniques, interdisciplinary collaboration, and ongoing innovation to ensure that LLM simulations remain reliable and ethically sound tools for political science research.

### C.4  LLM Explainability

**Definition of Explainability.** Explainability in the context of LLMs refers to the ability to provide interpretable and understandable outputs that clarify how and why specific predictions or decisions are made. In politically sensitive applications, explainability ensures that stakeholders can trace model outputs to underlying reasoning processes, fostering trust and transparency. Explainability is critical for validating insights derived from LLM analyses and ensuring fairness in decision-making for political science.

One of the ultimate goals of science is to *explain* phenomena and uncover cause-and-effect relationships. In political science, explainability plays a crucial role in understanding the impact of policies, campaigns, and social dynamics Feder et al. (2022); Ashwani et al. (2024); Zečević et al. (2023); Kıcıman et al. (2024). While explainability has been a focus in social science and medical research Feder et al. (2022), it has received comparatively less attention in political science Feder et al. (2022). LLMs, with their remarkable capabilities in language generation and pattern recognition, provide new tools for enhancing explainability in political tasks. However, they also face significant limitations in moving beyond correlation to meaningful reasoning Bagheri et al. (2024). These challenges hinder their ability to provide deeper explanations of the phenomena they analyze, an essential requirement for advancing scientific understanding. Despite these limitations, recent research highlights the potential of leveraging LLMs to enhance the explainability of political science applications, providing tools for researchers to explore cause-and-effect relationships in innovative ways.

**Explainability of LLMs in Political Science.** The explainability of LLMs, referring to the ability to generate interpretable insights, directly impacts their utility in causal inference Zhao et al. (2024).

Researchers can leverage explainability tools, such as attention mechanisms Luo & Specia (2024) and prompt engineering de Slegte et al. (2024), to identify relevant variables and interactions within data. For instance, post-hoc analysis methods Dhawan et al. (2024) enable researchers to interpret why an LLM has generated specific outputs, facilitating the identification of potential causal pathways in text-based datasets. This capability enhances the transparency and reliability of LLM-driven causal analysis, especially in politically sensitive contexts.

**Summary and Challenges.** LLMs hold significant promise for advancing explainability in political science by enabling researchers to interpret model decisions, analyze influential factors, and improve transparency in LLM-driven analyses. Their unique capabilities, such as leveraging attention mechanisms, generating rationale-based outputs, and providing structured justifications, make them valuable tools for enhancing interpretability. However, limitations such as biases, inconsistent explanations, and challenges in aligning model reasoning with human understanding must be addressed to ensure their effective application. As research continues, LLMs are poised to play an increasingly critical role of improving transparency, trust, and accountability in political science.

## C.5 Ethical Concerns in LLM Development and Deployment

**General Concerns About Embedded Values in LLMs.** Large language models are increasingly influencing societal and political discourse, raising fundamental questions about the values and biases they embed. The design and deployment of LLMs often involve implicit decisions about whose perspectives and moral frameworks are represented, potentially shaping public perception and decision-making in ways that are not always transparent. Johnson and Iziev Johnson & Iziev (2022) highlight the ethical dilemmas surrounding trust in AI-generated content, emphasizing the difficulty in ensuring that LLMs align with societal norms while avoiding the reinforcement of harmful biases. Similarly, Kim and Lee Kim & Lee (2023) examine the implications of LLM-driven conversational agents in political campaigns, noting the potential for these tools to inadvertently promote specific ideologies under the guise of neutrality. Lee et al. Lee et al. (2024c) further explore how LLMs reflect and propagate structural societal biases, particularly those affecting subordinate social groups. The study reveals that LLMs tend to portray these groups as more homogeneous, aligning with longstanding human cognitive biases, and underscores the importance of addressing such systemic issues in model training and evaluation. As LLMs continue to integrate into decision-making systems and public-facing applications, understanding their embedded values becomes imperative. This broad analysis sets the stage for more focused discussions on specific biases and potential mitigation strategies in subsequent sections.

**Specific Manifestations of Biases and Preferences in LLM Outputs.** The outputs of LLMs often reflect biases and preferences that manifest in specific, measurable ways, influencing how these models are perceived and utilized across different contexts. These manifestations not only reveal the underlying training data biases but also highlight the importance of careful model deployment. For instance, Tornberg Törnberg (2023) provides a comprehensive analysis of ChatGPT's language use, showing how the model tends to favor Western-centric cultural norms and professional jargon. This skew has implications for accessibility and inclusivity, as it may alienate users from non-Western backgrounds or those with varying levels of language proficiency. In addition, Stanczak et al. Stańczak et al. (2023) introduce a framework for quantifying biases in LLM outputs, with a focus on gender and occupational stereotypes. The study demonstrates that despite improvements in reducing overtly biased outputs, subtle biases persist, particularly in contexts where societal norms conflict with the training data distribution. Jiang et al. Jiang et al. (2022) also investigate how LLMs trained on community-specific data exhibit distinct preferences that align closely with the values and norms of those communities. While this approach can increase relevance for specific audiences, it raises concerns about the potential for reinforcing echo chambers and ideological polarization when these models are used in broader contexts. The findings collectively illustrate the challenges of mitigating biases in LLM outputs, calling for more robust evaluation mechanisms and the inclusion of diverse training data to minimize the risk of harmful stereotypes or cultural insensitivity.

**Practical Strategies for Mitigating Biases in LLMs.** Efforts to address the biases embedded in LLMs have led to the development of various practical strategies. These approaches aim to minimize the harm caused by biased outputs while maintaining the utility of the models in diverse contexts. Recent studies provide valuable insights into how such strategies can be implemented effectively. Rozado Rozado

(2023) emphasizes the importance of balancing ideological representations within LLMs to mitigate political biases. The study outlines a method of systematically curating training datasets to ensure parity in the representation of diverse viewpoints. This proactive approach not only reduces overt political biases but also fosters fairness in politically sensitive applications, such as journalism and policymaking. Building on this, Motoki et al. Motoki et al. (2024) highlight the role of iterative fine-tuning using diverse feedback sources. By incorporating user feedback from underrepresented communities, LLMs can better align with a broader range of cultural norms and values. The findings in Motoki et al. (2024) suggest that this dynamic feedback loop significantly enhances model responsiveness to marginalized perspectives, making it a crucial step in real-world deployments. Simmons Simmons (2023) takes a complementary approach by advocating for embedding explicit moral reasoning frameworks into LLM training pipelines. This strategy involves integrating ethical guidelines and decision-making frameworks into the model's architecture. Simmons argues that such measures not only mitigate biases but also equip models with the capacity to navigate morally ambiguous scenarios, thereby improving trustworthiness in high-stakes applications. These efforts demonstrate that mitigating biases in LLMs is both technically achievable and ethically essential.

**Broader Societal Implications of LLM Biases.** The biases embedded in LLMs extend beyond technical and academic concerns, influencing societal structures and interactions in profound ways. As LLMs become increasingly integrated into decision-making processes, communication platforms, and personalized services, understanding their broader societal impacts is critical. Scholar like Tornberg Törnberg (2023) highlights how biases in LLMs can perpetuate existing social inequalities by reinforcing dominant narratives. The study examines ChatGPT's performance in generating culturally sensitive responses, revealing disparities in the model's treatment of various sociocultural groups. Tornberg argues that such imbalances risk entrenching systemic inequities, especially when LLMs are used in education, public discourse, and policymaking. Alvarez et al. Alvarez et al. (2023) complement this analysis by exploring the role of generative AI in amplifying misinformation and political polarization. The study discusses how LLMs, if left unchecked, can contribute to the spread of ideologically skewed content, potentially exacerbating societal divisions. Alvarez emphasizes that biases in LLM outputs are not isolated technical flaws but are deeply intertwined with broader societal challenges, such as media manipulation and the erosion of public trust. Hackenburg and Margetts Hackenburg & Margetts (2024) extend these concerns to the realm of targeted advertising and political microtargeting. This study illustrates how biased LLMs can be leveraged to craft persuasive narratives tailored to specific demographics, raising ethical questions about manipulation and autonomy. Hackenburg warns that the misuse of biased language models in these contexts may deepen socioeconomic disparities and influence political outcomes in undemocratic ways. These studies highlight the importance of designing LLMs that are fair, transparent, and inclusive, particularly as they are increasingly applied in sensitive domains like political analysis and social sciences.

**Summary and Challenges.** The intersection of LLMs, societal values, and biases presents a complex but essential area of study. While advancements in LLMs enable transformative applications, their inherent biases pose significant ethical challenges. Addressing these challenges requires:

- *Awareness*: Achieving a deeper understanding of how biases manifest in LLM outputs.

- *Accountability*: Aligning LLMs with diverse societal needs under common ethical standards and guidelines.

- *Transparency*: Building methods for identifying, monitoring, and mitigating biases in real-world applications.

Future research must prioritize creating robust methodologies for bias mitigation, with a focus on enhancing fairness, inclusivity, and accountability in LLM development and deployment Motoki et al. (2024); Rozado (2023); Napolio (2024); Simmons (2023).

## C.6  Societal Impacts

**Definitions and Context.** The societal impacts of political-LLM sphere extend beyond technical concerns to encompass profound ethical, communicative, and informational implications. From influencing election outcomes to enhancing political communication, LLMs hold the potential to transform the societal landscape in

both positive and negative ways. This section explores the multifaceted effects of LLMs on political campaigns, public communication, and civic engagement, while addressing potential risks and ethical challenges.

**Transforming Political Campaigns.** LLMs have revolutionized the way political campaigns are conducted by enabling hyper-personalized messaging and voter targeting Bonikowski et al. (2022); Hackenburg & Margetts (2024); Moghimifar et al. (2024); Foos (2024); Yu et al. (2024c). Bonikowski et al. (2022) is an early work which highlights the potential of LLMs in measuring populism, nationalism, and authoritarianism through automated analysis of U.S. presidential debates. Hackenburg Hackenburg & Margetts (2024) demonstrates how LLMs can analyze large datasets to generate messages tailored to individual voter profiles, influencing voter perceptions and potentially altering election outcomes. Beyond voter engagement, LLMs play a strategic role in shaping campaign narratives that resonate with diverse audiences. Moghimifar et al. Moghimifar et al. (2024) show that LLM-based agents can model political coalition negotiations, providing insights into political alliances and enabling more dynamic campaign strategies. Foos Foos (2024) discusses how generative AI tools, including LLMs, are transforming election campaigns by facilitating AI-to-voter conversations and enabling scalable, multilingual interactions under diverse democracies. Lately, Yu et al. Yu et al. (2024c) propose a novel multi-step reasoning framework using LLMs for U.S. election predictions, incorporating time-sensitive factors like candidates' policies and demographic trends to enhance accuracy. Together, these works showcase the multifaceted capabilities of LLMs in modernizing political campaigns and amplifying their impact across various dimensions.

**Enhancing Political Communication** In an era of increasingly complex political discourses, LLMs offer tools to bridge the gap between policymakers and the public Argyle et al. (2023a); Alvarez et al. (2023); Gover (2023); Moghimifar et al. (2024); Ma et al. (2024). By simplifying intricate political and legislative content, LLMs make critical information more accessible to citizens, fostering greater political understanding and participation. Argyle et al. Argyle et al. (2023a) discuss how LLMs can distill party manifestos into understandable summaries, addressing barriers that often hinder public engagement. Similarly, Alvarez et al. Alvarez et al. (2023) highlight the potential of generative AI to enhance transparency and comprehension in elections, allowing voters to make more informed decisions. These advancements suggest that LLMs could play a pivotal role in democratizing information and improving the accessibility of political communication.

**Democratizing Information Access.** LLMs hold the promise of empowering individuals by breaking down complex topics into easily understandable language, thereby democratizing access to information. This capability can foster a more informed citizenry and enable greater accountability among political actors. By providing equitable access to political knowledge, LLMs ensure that more people, regardless of educational background, can participate in democratic processes. For instance, LLMs can assist in translating political jargon or simplifying policy discussions, helping individuals navigate traditionally opaque political systems. This democratization of information will lead to a more inclusive political landscape.

**Ethical Risks.** While LLMs offer substantial benefits, their societal deployment also raises critical ethical concerns. One major issue is the potential misuse of LLMs to disseminate misinformation or biased content, which could manipulate public opinion or destabilize democratic processes. Bai et al. Bai et al. (2023) discuss the persuasive power of LLM-generated text in influencing political opinions, underscoring the need for safeguards to mitigate risks. Furthermore, the ability of LLMs to generate realistic but misleading content poses challenges in distinguishing fact from fiction, creating vulnerabilities for misinformation campaigns. Addressing these ethical challenges require robust governance frameworks and continuous monitoring.

**Summary and Challenges.** The societal impacts of LLMs are vast and multifaceted, offering opportunities to enhance political communication while raising ethical and democratic concerns. To fully leverage the potential of LLMs while mitigating risks, future research and governance efforts must focus on:

- *Accountable Deployment*: Establishing guidelines for the ethical use of LLMs in politically sensitive contexts.

- *Transparency*: Developing tools to track and explain LLM-generated content to avoid misuse.

- *Public Awareness*: Educating users about the benefits and potential risks of LLMs to promote informed and responsible decision-making.

- *Misinformation Prevention*: Implementing safeguards to detect and counteract biased or false narratives.

By addressing these challenges, LLMs can contribute to a more equitable and transparent political environment, ensuring their societal impacts remain positive.

## D  Datasets in Computational Political Science: Benchmarks, Insights, and Preparation Strategies

### D.1  Benchmark Datasets

To meet the specific demands of political science applications, various benchmark datasets grounded in real-world data have been developed to evaluate LLMs on tasks such as sentiment analysis, election prediction, legislative summarization, misinformation detection, and conflict resolution. Each dataset is designed with domain-specific criteria to assess the alignment of LLM outputs with real-world political and social contexts, ensuring their relevance and applicability to practical scenarios. A comprehensive list of these datasets, along with their respective tasks and characteristics, is presented in Table 3 to facilitate reference and comparison.

**Sentiment Analysis & Public Opinion Dataset.** Various datasets have been developed to accurately assess LLMs in sentiment analysis and public opinion. For instance, OpinionQA Santurkar et al. (2023) is designed as a test environment where LLMs answer questions about public opinion, capturing subtle sentiments across 1,489 well-crafted queries. This dataset is valuable because it benchmarks how closely LLMs can align with actual human opinion patterns—a key factor for extracting sentiment accurately in social sciences. Similarly, PerSenT Bastan et al. (2020) focuses on tracking sentiments toward specific entities mentioned in news articles. It tests how well LLMs can detect and follow opinions expressed by particular individuals, allowing for sentiment to be aggregated over multiple mentions of popular entities to support comprehensive public opinion analysis. In addition, GermEval-2017 Chebolu et al. (2022) provides a corpus of social media comments about Deutsche Bahn, the railway service in Germany, tailored for aspect-based sentiment analysis. This would help organizations and service providers derive actionable insights from feedback by homing in on specific aspects such as noise levels or punctuality. Datasets like Twitter Sharma et al. (2022), Bengali News Comments Saha et al. (2022), and Indonesia News Waspodo et al. (2022) extend the sentiment analysis to widely used social and news media platforms in multiple languages. These multilingual datasets are significant for cross-linguistic and cultural sentiment studies, which find especially relevant applications in global social media and market research.

**Election Prediction & Voting Behavior Dataset.** The U.S. Senate Statewide 1976-2020 MIT Election Data and Science Lab (2017c) dataset contains state-level election returns, while the U.S. House 1976-2022 MIT Election Data and Science Lab (2017b) dataset provides district-level returns, offering resources for analyzing nearly five decades of electoral trends. Other than that, The U.S. Senate Returns 2020 MIT Election Data and Science Lab (2022a) and U.S. House Returns 2018 MIT Election Data and Science Lab (2022b) datasets offer detailed precinct-level voting data, allowing LLMs to analyze U.S. voting patterns and voter behavior with the highest granularity, which supports election prediction and voting behavior studies. The State Precinct-Level Returns 2018 dataset MIT Election Data and Science Lab (2022c), with its extensive 10 million data points, provides a substantial resource for LLMs to train on and analyze voting behaviors comprehensively. The 2008 American National Election Study (ANES) Payne et al. (2010) offers insights into voter preferences and political attitudes through surveys conducted before and after the election, capturing differences in voter sentiment, which LLMs can model to reflect public opinion changes. The U.S. President 1976–2020 dataset MIT Election Data and Science Lab (2017a) provides historical data essential for LLMs to examine long-term political trends and election outcomes across multiple decades. These datasets serve as invaluable training sources for LLMs to support political campaigns, media analysis, and social science research into electoral behaviors and trends.

**Legislation & Administrative Rules Dataset.** For summarizing and analyzing legislation and administrative rules, key datasets include BillSum Kornilova & Eidelman (2019), CaseLaw Shu et al. (2024) and Federal Register Moore (2018). BillSum aims to offer support to summarize US Congressional bills; it empowers LLMs to process mid-length legislative text and to produce brief summaries, which would

**Table 3:** Existing benchmark datasets in LLM for Political Sciences.

| Benchmark Datasets | Application Domain | Evaluation Criteria |
|---|---|---|
| OpinionQA DatasetSanturkar et al. (2023) | Sentiment Analysis & Public Opinion | Ability to answer 1,489 questions |
| PerSenTBastan et al. (2020) | Sentiment Analysis & Public Opinion | Performance on 38,000 annotated paragraphs |
| GermEval-2017Chebolu et al. (2022) | Sentiment Analysis & Public Opinion | Accuracy on 26,000 annotated documents |
| TwitterSharma et al. (2022) | Sentiment Analysis & Public Opinion | Analysis of 5,802 annotated tweets |
| Bengali News CommentsSaha et al. (2022) | Sentiment Analysis & Public Opinion | Performance on 13,802 Bengali news texts |
| Indonesia NewsWaspodo et al. (2022) | Sentiment Analysis & Public Opinion | Sentiment analysis on 18,810 news headlines |
| U.S. Senate Statewide 1976-2020 MIT Election Data and Science Lab (2017c) | Election Prediction & Voting Behavior | Analysis of 3,629 data points |
| U.S. House 1976-2022 MIT Election Data and Science Lab (2017b) | Election Prediction & Voting Behavior | Analysis of 32,452 data points |
| U.S. Senate Returns 2020MIT Election Data and Science Lab (2022a) | Election Prediction & Voting Behavior | Prediction accuracy on 759,381 data points |
| U.S. House Returns 2018MIT Election Data and Science Lab (2022b) | Election Prediction & Voting Behavior | Analysis of 836,425 data points |
| State Precinct-Level Returns 2018MIT Election Data and Science Lab (2022c) | Election Prediction & Voting Behavior | Analysis of 10,527,463 data points |
| 2008 ANES Time Series StudyPayne et al. (2010) | Election Prediction & Voting Behavior | Analysis of 2,322 pre-election and 2,102 post-election surveys |
| 2016 ANES Time Series StudyYu et al. (2024c) | Election Prediction & Voting Behavior | Analysis of 2,322 pre-election and 2,102 post-election surveys |
| U.S. President 1976–2020MIT Election Data and Science Lab (2017a) | Election Prediction & Voting Behavior | Analysis of 4,288 data points |
| BillSumKornilova & Eidelman (2019) | Legislation & Administrative Rules | Summarization of 33,422 U.S. Congressional bills |
| CaseLawShu et al. (2024) | Legislation & Administrative Rules | Analysis of 6,930,777 state and federal cases |
| DEU IIIArregui & Perarnaud (2022) | Legislation & Administrative Rules | Performance on 141 legislative proposals and 363 controversial issues |
| Federal Register Final Rule Data 2000-2014Moore (2018) | Legislation & Administrative Rules | Titles and Summaries of 61,216 U.S. Federal Regulations |
| PolitiFactShu et al. (2020) | Misinformation Detection | Detection across six integrated datasets |
| GossipCopGrover et al. (2022) | Misinformation Detection | Detection across ten integrated datasets |
| WeiboJin et al. (2017) | Misinformation Detection | Classification of 4,488 fake news and 4,640 real news items |
| SciNewsCao et al. (2024) | Misinformation Detection | Detection in 2,400 scientific news stories |
| UCDPCunningham et al. (2013) | Game Theory & Negotiation | Analysis of armed conflicts and peace agreements |
| PNCCArı (2023) | Game Theory & Negotiation | Data on peace agreements and conflict resolution |
| WebDiplomacyFundamental AI Research Diplomacy Team et al. (2022) | Game Theory & Negotiation | Analyze 12,901,662 messages exchanged between players |

considerably reduce the efforts of experts from the legal community and policy analysis. The CaseLaw dataset provides an extensive collection of state and federal cases, serving as a foundation for LLMs to analyze legal precedents and support judicial decision-making. The DEU III dataset Arregui & Perarnaud (2022) spans three decades of EU legislative decision-making, enabling the evaluation of LLMs in analyzing policy positions and negotiation dynamics among EU member states and institutions. Beyond legislation, the U.S. Federal Register dataset Moore (2018) includes titles and summaries of all final federal rules from 2000 to 2014, focusing on administrative decisions. This dataset provides a valuable resource for LLMs to analyze regulatory trends and the decision-making processes of federal agencies.

**Misinformation Detection Dataset.** To address the negative effects of fake news and misleading information, several open-sourced datasets have been constructed Grover et al. (2022); Jin et al. (2017). PoliFact Shu et al. (2020) supports the use of large language models to distinguish between false and genuine news by focusing on publisher behavior, user interactions, and network structures. Similarly, SciNews Cao et al. (2024) concentrates on misinformation in scientific reporting, providing a resource that helps preserve the integrity of science communication and limit the spread of misleading health and science information.

**Game Theory & Negotiation Dataset.** In the domain of conflict resolution and game theory research, there are datasets that guide the study of strategic interactions and peace negotiations. For example, the Non-State Actors in Armed Conflict (NSA) dataset Cunningham et al. (2013) includes information on state-rebel group dyads, enabling more detailed examinations of conflicts with actor-specific data. In addition, the Peace Negotiations in Civil Conflicts (PNCC) dataset Arı (2023) documents formal negotiation phases during civil conflicts. Moreover, the WebDiplomacy dataset Fundamental AI Research Diplomacy Team et al. (2022) consists of message exchanges between players in a simulated diplomatic negotiation setting, enabling a clearer understanding of communication patterns and strategic decision-making in conflict scenarios.

These benchmark datasets, taken together, provide a solid mainstay for a truly large number of LLMs applications in political science, from voter sentiment analysis to the exploration of legislative choices, tracking misinformation, and modeling conflict negotiations.

## D.2   Dataset Preparation Strategies

Dataset preparation is a critical step in adapting LLMs for downstream political science applications Yu et al. (2024d). Given that the adaptation of LLMs in computational political science (CPS) is still in its infancy, the publicly available benchmark datasets remain scarce. The preparation of CPS datasets requires careful consideration of both domain-specific and generalizable strategies Lin (2024); Wagner et al. (2024). Drawing insights from adjacent research fields like general sentiment analysis, fake news detection, and LLM-based dialogue generation, political datasets can be adapted to align with tasks such as election prediction, policy analysis, and political discourse generation.

**Broad Source of Dataset Collection.** One primary approach of dataset preparation involves collecting text data from publicly available political sources, such as speeches, legislative records, news articles, and social media platforms. For instance, in OpinionQA Santurkar et al. (2023) and PerSenT Bastan et al. (2020), the data is sourced from political discussions and news media, which is then annotated for tasks like opinion alignment and sentiment detection. To ensure the data is relevant and representative, these dataset collections usually focus on specific political events, ideologies, or actors, which are essential for training LLMs to understand political discourse.

OpinionQA Dataset Generation

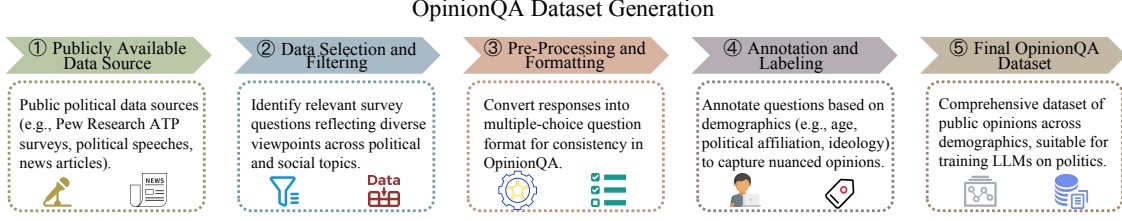

**Figure 7:** Illustration of the OpinionQA dataset preparation on publicly available data source.

We elaborate the developing process of OpinionQA dataset in Figure 7. To start with, researchers utilized publicly available data from various political and social surveys as the source data. They particularly leverage Pew Research's American Trends Panel (ATP) surveys, which span a wide array of topics, including science, politics, and social issues. The dataset compilation process involves selecting pertinent survey questions that reflect diverse viewpoints across key issues and topics in the United States. These survey responses are preprocessed to create a multiple-choice question format, which serves as a reliable structure for language models to interpret. Through the methodology, each question in OpinionQA is annotated based on survey results, representing public opinion across various demographics such as age, political affiliation, income, and ideology. This approach ensures that the dataset encapsulates the complexity and nuance of real-world opinions, which are essential for training language models to simulate and interpret politically charged discourse accurately.

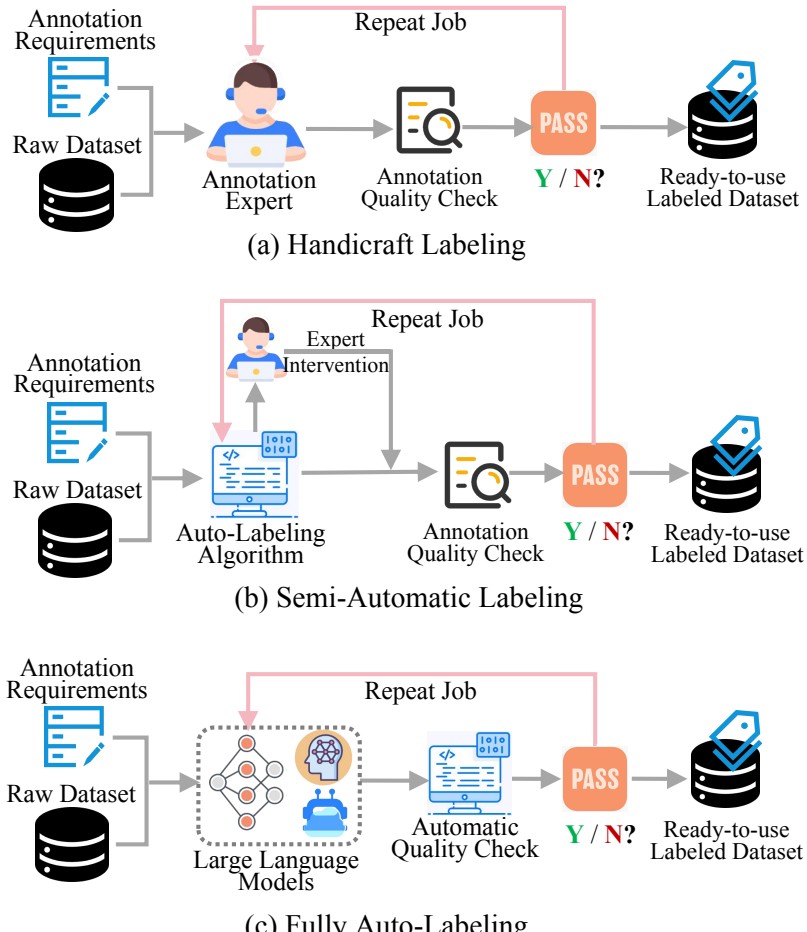

**Figure 8:** Illustration of dataset annotation approaches, including traditional manual approach, semi-automated approach, and LLM-based fully automated approach.

**Annotation Strategies.** Annotation is another essential aspect of dataset preparation. Datasets intended for political sentiment analysis or misinformation detection require detailed labeling, often involving either expert or crowd-sourced annotations Mochtak et al. (2023). For instance, the State Precinct-Level Returns 2018 dataset MIT Election Data and Science Lab (2022c) includes a substantial amount of real, unannotated data. Training LLMs with such data may involve adding annotations to capture sentiment toward political entities or identify media biases. Annotation schemes should be crafted to reflect nuanced political ideologies and opinions, ensuring that the dataset reflects the diversity and complexity of political discourse Balloccu et al. (2024); Rauniyar et al. (2023).

As shown in Figure 8, annotation can be conducted through different approaches. These methods range from fully manual labeling Tan et al. (2024), where annotation experts review and label the data by hand, to semi-automated processes that use algorithms to assist with labeling Huang et al. (2024), with experts intervening as needed. In fully automated labeling, LLMs or other automated systems can handle the labeling work entirely, followed by a quality check Ming et al. (2024). Each method has its trade-offs among accuracy, scalability, and manual effort required.

**Dataset Bias and Representation.** Addressing bias and representation is particularly crucial in political science datasets. Datasets must account for the diversity of political systems, ideologies, and demographics Qu & Wang (2024); Shahbazi et al. (2023). Researchers must ensure the collected political datasets are balanced across different viewpoints and the data does not over-represent certain political ideologies. Techniques such as oversampling underrepresented groups or creating synthetic data using LLMs can be employed to achieve this balance Nakada et al. (2024); Cloutier & Japkowicz (2023).

**Data Preprocessing & Normalization.** Given the complexity of political language, appropriate preprocessing and normalization are indispensable Chai (2023). Preprocessing steps such as entity recognition, text cleaning, and the extraction of key political terms help standardize the input and improve the model's ability to learn from diverse contexts of political science Ehrmann et al. (2023). These techniques ensure that LLMs can process the input text effectively.

**Data Augmentation.** Augmentation strategies like paraphrasing or generating synthetic data with LLMs help to expand the dataset size in cases where political data is limited Sahu et al. (2023); Abaskohi et al. (2023). Data augmentation helps diversify the training set, allowing the model to generalize better to new and unseen political scenarios dos Santos et al. (2024); Ding et al. (2024).

To further illustrate how these strategies applied to practical scenarios, we now introduce three examples of dataset preparation tailored for specific LLM-based political science tasks. Each example demonstrates how researchers effectively leverage LLMs to address key challenges in political data curation and annotation:

**(1) Developing a Dataset for LLM-Based Political Debiasing.** For the political debiasing task, constructing a dataset involves curating a balanced collection of political texts that represent diverse political ideologies and viewpoints. For instance, to debias LLM outputs, we can gather news articles, social media posts, and political speeches from various political parties, regions, and ideologies. The dataset will need to be annotated with the political bias present in each text. This can be done using a combination of manual annotation by political experts and automated tools to identify biased language, sentiment, and framing. The goal is to provide a dataset that allows the model to recognize and mitigate its inherent biases by learning from a balanced set of inputs across the political spectrum.

**(2) Automated Annotation Using LLMs: Example in Legislative Interpretation.** LLM-based legislative interpretation is a promising application in political science. Using a dataset like BillSum Kornilova & Eidelman (2019), which includes U.S. legislative documents, LLMs can be employed to automatically annotate sections of the legislation with relevant policy categories, key provisions, and political implications. LLMs can also be fine-tuned on a smaller, manually annotated set of legislative texts in order to classify different legal concepts and policy issues. This automated annotation streamline will accelerate the process of categorizing large volumes of legislative content, helping political analysts and lawmakers quickly interpret and summarize complex bills.

**(3) Generating Synthetic Political Datasets Using LLMs.** The limitations in acquiring large and diverse political datasets due to privacy, restrictions, and sensitivities make generating synthetic datasets with LLMs a promising solution. Considering election prediction as an example, LLMs are able to generate hypothetical voter opinion surveys based on historical election data and known demographic trends. By training LLMs on existing public opinion survey datasets, researchers can generate synthetic datasets that simulate different electoral conditions, voter behaviors, and political trends. This approach will greatly enhance the availability of diverse political data for training and testing election prediction models.

