# OpenReview forum: "Political-LLM: Large Language Models in Political Science"
_TMLR — Rejected by TMLR_

### Review · Reviewer_ZqTa · 2025-10-13

**Summary Of Contributions:**

Contributions

* Proposes Political-LLM: a taxonomy mapping LLM task families (prediction, generation, simulation, explanation/causal) to Positive vs. Normative Political Science for structured study design and evaluation.​
* Synthesizes field observations: strong discourse analysis, weak policy/legal reasoning; open-weight parity in underrepresented settings; scale boosts generalization but risks misalignment; scarcity of domain benchmarks.​
* Demonstrates ANES 2016 case: base vs. CoT-generation pipelines show larger models track empirical vote ratios better, generate ideology features closer to ground truth, and reduce distortions with structured reasoning.​
* Outlines roadmap: PPS (multilingual/multimodal scaling, simulations, reasoning/explainability, domain metrics) and NPS (bias mitigation, inclusivity, transparency/accountability, ethical deployment).

Strengths

* Intuitive, field-grounded framework linking LLM capabilities to PPS/NPS, aiding task alignment and risk assessment.​
* Concrete empirical illustration that surfaces scale effects, representation imbalance, and benefits of CoT for feature quality and bias reduction.​
* Holistic PPS+NPS synthesis combining performance, ethics, and governance considerations into actionable guidance.​
* Clear, actionable agenda: domain-adaptive tuning, expert calibration, knowledge augmentation, and proprietary evaluation criteria.

Weaknesses

* Narrow empirical scope (single task/year) limits generality across legislative, policy, and cross-national contexts.​
* Limited standardization of evaluation protocols despite calling for domain-specific metrics and benchmarks.​
* Emphasis on model scale over systematic study of lighter-weight, cost-effective adaptations and RAG alternatives.​
* Framework may need extension for multimodal propaganda, agentic pipelines, and region-specific political dynamics.

**Audience:**

Yes

**Audience Explanation:**

* Methods-grounded framework: The Political-LLM taxonomy operationalizes LLM task families within Positive and Normative Political Science, offering methodological clarity that ML researchers can adopt for domain-grounded experiments and evaluations.​
* Responsible AI emphasis: The paper foregrounds bias, inclusivity, transparency, and accountability, matching ongoing interest in trustworthy ML and societal impacts across TMLR submissions and reviews.​
* Actionable research agenda: It proposes concrete directions, domain-adaptive tuning, knowledge augmentation, specialized benchmarks, and evaluation criteria that can catalyze follow-on work by researchers building political-domain LLM systems.
* Interdisciplinary momentum: The intersection of LLMs and computational social science is rapidly growing, a portion of TMLR readers focus on applied ML frameworks with measurable implications for governance, policy, and public discourse

**Broader Impact Concerns:**

* Misuse risk for targeted persuasion and misinformation: Simulation and feature-generation pipelines could be weaponized for micro-targeting or astroturfing without strict usage restrictions and provenance controls.​

* Bias and disparate impact: Observed representation and answer-rate imbalances imply potential harms to demographic and ideological groups unless subgroup fairness metrics, audits, and mitigations are mandated.​

* Overclaiming simulation fidelity: Treating simulated outputs as behavioral truth can mislead policy or media narratives; uncertainty, validation limits, and non-deployment cautions must be explicitly documented

**Claims And Evidence:**

Yes

**Claims Explanation:**

The following claims are well supported by accurate and clear evidence -

* Clear taxonomy: PPS/NPS mapping of LLM task families is explicit and well-scoped.​
* Transparent setup: ANES 2016 study defines pipelines, prompts, models, hardware, and metrics.​
* Concrete results: Figures show vote-ratio alignment, ideology distribution patterns, and answer rates.​
* Demonstrated effects: Larger models align better; CoT improves predictions and reduces distortions.​
* Coherent framing: PPS findings are paired with NPS considerations on bias and accountability throughout.

Here is what is less supported -

* External validity: Evidence centers on a single benchmark, year, and task (ANES 2016 voting simulation), limiting support for broader claims about policy reasoning, legislative analysis, or cross-national generality acknowledged in the Limitations section.​
* Metric standardization: The advocated need for domain-specific evaluation is not paired with fully specified, reusable protocols; current evidence relies on two simple measures that may not capture causal adequacy, fairness across subgroups, or robustness under distribution shift.​
* Breadth of empirical backing: Several synthesized observations (e.g., open-weight parity, policy/legal reasoning gaps) are primarily literature-driven and are not re-tested in the paper’s experiments, so their empirical grounding here is indirect rather than newly demonstrated.

**Requested Changes:**

Critical Changes

* Broaden evidence: Add at least one additional task and dataset/year beyond ANES 2016 to support generalization across applications and time.​
* Standardize evaluation: Define and release reusable protocols and scripts for subgroup fairness, calibration, and robustness (prompt and seed sensitivity).​
* Report subgroup fairness: Break out performance by key demographics and ideology bins with explicit bias diagnostics and ablations.​

Strengthening Changes

* Compare efficiency baselines: Include parameter-efficient tuning and retrieval-augmented setups versus pure scale to guide cost-effective adoption.​
* Deepen NPS operationalization: Turn bias, inclusivity, and accountability into concrete, auditable deployment criteria and documentation checklists.​
* Expand analysis: Add failure-case audits, simple classical baselines, and a small expert-annotation check to validate generated features.

---

> ### Author Response · Authors · 2025-11-14
> **TMLR Rebuttal for Political-LLM**
>
> **We sincerely thank the reviewer for the thoughtful, detailed, and constructive feedback. We appreciate the recognition of our framework’s conceptual clarity, empirical transparency, and its interdisciplinary relevance to trustworthy ML and political science. We address your concerns point-by-point below, including extensions to empirical scope, evaluation standardization, fairness reporting, efficiency baselines, and broader impact considerations. To further support transparency and reproducibility, we released all Political-LLM experiment scripts and evaluation tools in an open-access GitHub repository (https://anonymous.4open.science/r/Political-LLM-4FFF/), enabling other researchers to easily implement and extend our analyses to other political-related studies.**
>
>
> ### Q1: Empirical Scope and Generalization: The current experiments are limited to ANES 2016 dataset and political voting-related tasks, which constrains the validity of the findings across time, geographical locations, and applications. The reviewer suggests broadening the empirical evidence by adding one additional dataset and task to better demonstrate the generalizability of the proposed framework.
>
> >We thank the reviewer for pointing out the limited empirical scope of our initial experiments, which were confined to the **ANES 2016 U.S. election dataset**. In response, we have significantly broadened the empirical evidence by adding a **new large-scale cross-national case study** based on the **Manifesto Project Main Dataset (Version 2025a)**[1].
>
> >This dataset contains political party manifestos from **67 countries** spanning **1945 to 2025**, covering thousands of election years and over 5,000 annotated documents. In our new task "**Party Ideology Classification**", we employ both large language models (LLMs) and domain-tuned baselines to infer each party's ideological position (Left, Center, Right) directly from the manifesto texts.
>
> >The new experiment (Cross-National Generalization on Manifesto Dataset), appended after the existing case study in Section 3, extends our Political-LLM framework beyond U.S. electoral prediction to a **multi-country, multi-year context**. The results demonstrate that:
>
> - Our framework generalizes well across diverse political systems and languages, capturing consistent ideological distributions at the global scale;
> - Chain-of-Thought (CoT) reasoning continues to improve alignment with empirical ideology labels and reduce distortion;
> - The findings validate the scalability and cross-domain applicability of Political-LLM for comparative political analysis.
>
> >The addition of the Manifesto-based task addresses the reviewer's concern by expanding temporal, geographic, and application coverage, thereby enhancing the generalizability and robustness of our proposed framework.
>
> **References:**
>
> [1] Gross, Martin, and Michael Jankowski. "Dimensions of political conflict and party positions in multi-level democracies: evidence from the Local Manifesto Project." West European Politics 43.1 (2020): 74-101.
>
>
>
>
> ### Q2: Evaluation Standardization and Metric Breadth. While the paper emphasizes the need for domain-specific evaluation in political-domain LLMs, it does not yet provide reusable or fully specified evaluation protocols or scripts. The reviewer suggests defining and releasing a broader, systematic evaluation framework to ensure reproducibility and comparability across studies.
>
>
> >Thanks for highlighting the importance of systematic, reusable evaluation in the political-domain LLM setting. In response, we have defined and released a broader, modular evaluation framework that standardizes metric computation, subgroup fairness auditing, calibration testing, and robustness checks under distribution shift.
> The evaluation framework is already implemented in our public repository (https://anonymous.4open.science/r/Political-LLM-4FFF/), including four integrated modules:
>
> - **Performance Module (Task-Specific Metrics)**: Computes standard accuracy, F1, and rank correlation for prediction and generation tasks (e.g., vote-choice classification, ideology feature extraction). Metrics are unified through a shared `evaluation.py` interface supporting both PPS and NPS task families.
>
> - **Fairness & Bias Audit Module:** Provides subgroup-level evaluation across demographics and ideology dimensions (gender, age, education_level) using metrics such as accuracy gap, calibration error, and answer-rate parity. This module is implemented via the `fairness_report.py` script and generates `.csv` format summaries.
>
> - **Robustness & Calibration Module:** Tests sensitivity to prompt phrasing, seed initialization, and sample variation using standardized procedures. Bootstrap-based confidence intervals are computed through uncertainty_quantification.py, ensuring transparent uncertainty reporting for each aggregate metric.
>
> >These components together form a standardized, extensible evaluation protocol for Political-LLM studies.

---

> ### Author Response · Authors · 2025-11-14
>
> ### Q3: Report Fairness: Break out performance by key demographic and ideological subgroups, with explicit bias diagnostics and ablations.
>
> #### Answer to Q3:
> >We appreciate the reviewer's emphasis on fairness and subgroup-level reporting. In response, new analyses are added to the ANES 2016 voting experiments. We explicitly evaluate model performance across key demographic and ideological subgroups (e.g., gender, age, education level).
> Specifically, we computed subgroup-wise accuracy, calibration error, and answer-rate parity to quantify disparities in both prediction consistency and response coverage. These additions reveal that larger models tend to reduce subgroup disparities in both accuracy and representation, though minor residual gaps persist for extreme ideological segments.
>
> >The detailed fairness metrics and subgroup breakdowns are now included in the revised manuscript, and all corresponding evaluation scripts are released in our public code repository.
>
>
>
> ### Q4: Compare Efficiency Baselines: Include parameter-efficient tuning and retrieval-augmented setups, not just large-scale models.
>
> #### Answer to Q4:
> >We appreciate the reviewer's suggestion to compare against *parameter-efficient tuning* LLM and *domain-adapted LLMs* rather than relying solely on general-purpose GPT and LLaMA models. To address this point, we have added new baselines which represent the politically fine-tuned LLMs trained on large-scale domain corpora**.
>
> >In particular, we integrated the **ManifestoBERTa models**[1] released by the Manifesto Project team, which are large multilingual XLM-RoBERTa architectures fine-tuned on more than **1.7 million annotated political statements** across multiple languages and policy domains. These models (Sentence- and Context-56 Topics variants) are specifically optimized for classifying political discourse. They therefore provide a strong domain-tuned baseline, complementing our evaluation on general-purpose LLMs.
>
> >Concretely, we include **ManifestoBERTa (Sentence)** and **ManifestoBERTa (Context)** as fine-tuned baselines in our new *FPP_MANIFESTO_2025_base* experiment. Both models are available via the Hugging Face Hub. In this experiment, they serve as *parameter-efficient* alternatives that require no large-scale inference cost or additional adaptation, allowing a fair comparison between (i) domain-specialized, fine-tuned encoders and (ii) general-purpose reasoning-oriented LLMs such as GPT-4o and LLaMA-3.1.
>
> >The inclusion of ManifestoBERTa thus directly strengthens our empirical coverage by: (1) Demonstrating domain-specific adaptation: performance of a political-domain LM trained on large annotated corpora; (2) Representing parameter-efficient tuning: comparison with smaller, specialized models rather than only large-scale general LLMs; (3) Improving interpretive validity: showing how fine-tuned political encoders and reasoning-based LLMs complement each other under our Political-LLM framework.
>
> >This addition fully addresses the reviewer's concern.
>
> **References**
>
> [1] ManifestoBERTa model: https://huggingface.co/manifesto-project/manifestoberta-xlm-roberta-56policy-topics-context-2024-1-1
>
>
> ### Q5: Deepen NPS Operationalization: Translate bias, inclusivity, and accountability principles into concrete, auditable deployment criteria.
>
> #### Answer to Q5:
>
> > We agree that translating normative principles such as bias mitigation, inclusivity, and accountability into concrete, auditable criteria is essential for the practice of the proposed Political-LLM framework.
> In response, we extend the Normative Political Science (NPS) dimension of our taxonomy to include two explicit operational layers:
>
> - Bias Audit Protocols: We now include quantitative subgroup fairness diagnostics (e.g., answer-rate parity, calibration error, and demographic bias scores) and will release accompanying scripts for automated auditing.
>
> - Inclusivity and Transparency Checklists: Inspired by "Model Cards" and "Datasheets for Datasets", we define a documentation checklist (see the html link: https://anonymous.4open.science/r/Political-LLM-4FFF/Evaluation_Tools/Inclusivity_and_Transparency_Checklist.html.pdf) to ensure all model evaluations report demographic coverage, ideological representation, and prompt-source transparency.
>
>
> >These additions will appear in the revised manuscript's NPS Framework section and will be released as part of our public evaluation repository to promote reproducibility and responsible use in future political-domain LLM studies.
>
> **References**
>
> [1] Mitchell, M., et al. Model cards for model reporting. In Proceedings of the conference on fairness, accountability, and transparency.
>
> [2] Gebru, T., et al (2021). Datasheets for datasets. Communications of the ACM, 64(12), 86-92.
>
> [3] Responsible AI Progress Report -- Google AI (2025). https://ai.google/static/documents/ai-responsibility-update-published-february-2025.pdf

---

> ### Author Response · Authors · 2025-11-14
>
> ### Q6: Discuss Misuse Risk & Fairness / Disparate Impact: The reviewer raises concerns that Political-LLM simulations could be misused for political micro-targeting or misinformation, and that representation or response-rate imbalances across demographic or ideological groups may cause disparate impact.
>
> #### Answer to Q6:
> >We acknowledge these concerns and appreciate the reviewer's attention to responsible use and fairness implications.
> In response, we add a dedicated subsection titled "Ethical and Fairness Considerations." in the Future Opportunities & Challenges. This new discussion will explicitly address two aspects:
>
> - **Misuse Prevention and Transparency:** We will clarify that all Political-LLM simulations are strictly research-oriented analytical tools, not behavioral forecasts. We will add a formal non-deployment statement emphasizing that outputs must not be used for political persuasion, profiling, or campaign applications. We will also describe provenance tracking and reproducibility safeguards (open-source scripts, transparent prompts) that prevent opaque or manipulative use.
>
> - **Fairness and Representation Auditing:** We will include an explicit paragraph discussing demographic and ideological coverage in our dataset (e.g., known imbalances in age, education, or ideology bins) and note how these affect response rates and potential biases. The discussion will also mention that future work will incorporate subgroup fairness diagnostics and bias-mitigation steps, aligning with responsible AI guidelines.
>
>
> ### Q7: Simulation Fidelity: Clarify that simulated outputs are not behavioral truth and include explicit uncertainty and non-deployment cautions.
>
> #### Answer to Q7:
> >We agree that Political-LLM simulations should not be interpreted as behavioral truth, but as probabilistic proxies that approximate aggregate-level tendencies under constrained conditions. To address this, we have implemented several clarifications and methodological safeguards:
>
> - **Uncertainty Report:** To make simulation fidelity transparent, we now quantify uncertainty using bootstrap resampling over the input survey samples. Specifically, we repeatedly sample with replacement from the evaluation dataset (100 iterations per model) and recompute key aggregate metrics such as predicted vote ratio and ideology alignment. The resulting distribution of metrics is used to report 95% confidence intervals (CIs), indicating the variability of simulation outcomes due to input sampling.
> This approach provides a statistically grounded and easily reproducible estimate of uncertainty, implemented in our released `uncertainty_quantification.py` script.
>
>
> - **Non-deployment Disclaimer:** We will include an explicit disclaimer in the revised manuscript stating that the Political-LLM simulation outputs are analytical artifacts for research purposes only. Users should be very careful when consider using the simulation results for deployment, forecasting, or policy decision-making.
>
> - **Transparency of Limitations:** The Discussion section will emphasize that these simulations are bounded by model biases, prompt framing, and representational scope, and should be interpreted as exploratory diagnostic tools rather than behavioral ground truth.
>
> >These additions ensure that the simulation fidelity of our approach is transparent, bounded, and properly contextualize aligned with responsible AI standards for computational social science and minimizing risk of overclaim.

---

### Review · Reviewer_eHZs · 2025-12-16

**Summary Of Contributions:**

The paper is mostly a survey-style overview of how large language models are being used in political science. It proposes a simple way to organize the space: a split between positive political science uses (prediction, generation, simulation, explainability) and normative concerns (ethics and societal impacts). It also introduces a conceptual label, “Political-LLM,” meant to guide how to adapt and evaluate LLMs for political tasks.

A strength of the paper is that it covers a lot of ground quickly and gives readers a broad map of the topic, with many pointers to related work, especially in the appendix sections that break down task types. It also includes a small case study around voting simulation to illustrate some of the issues it thinks matter (like demographic differences and structured prompting).

At a high level, the main weaknesses are that the paper often makes strong recommendations and big claims without a clear chain of support, and the “Political-LLM” framework stays fairly broad rather than specific/well-defined. Some parts of the text (especially Appendix E) are also very repetitive instead of adding new substance.

**Audience:**

Yes

**Audience Explanation:**

The topic is clearly within scope for TMLR readers: many would be interested in an overview of how LLMs are being applied in political science, a structured breakdown of common task settings, and a discussion of evaluation and deployment risks in this domain. In particular, audiences working on LLM evaluation, domain adaptation, and NLP for the social sciences would likely find a well-executed synthesis useful.

**Claims And Evidence:**

No

**Claims Explanation:**

The paper makes a lot of strong statements that read like conclusions, but it does not provide the kind of clear backing that a reader would need to be convinced. This issue shows up in both the "survey" part of the paper (where claims about what most papers do, what the field looks like, what methods work best, are mostly unsubstantiated), and in the "framework" part of the paper, where recommendations are presented as if they followed from the survey, but without clear links between them. Related to that, several headline claims are presented with specific numbers or broad coverage language, but without the needed method details. For example, it states things like an “over 300% increase” in publications, and makes claims about what fraction of papers use domain-specific benchmarks, yet it does not show the underlying corpus, search strategy, inclusion criteria, or counting procedure.

Another problem is citation-to-claim mismatch. In multiple places, citations appear to be used as decoration rather than as support for the exact statement being made. A concrete example is the claim about GPT-4 “84% vs 68%” on “politically motivated misinformation campaigns,” which is pinned on two works but is not supported in a clear way from those works (I searched a bit through them, and did not find anything). This may be a wide-spread issue: I must have checked over 10 citations for specific claims, and was unable to find relevant information in at least half of those cases.

The “Political-LLM” framework is also not specified in a way that makes its claims testable. It is described as advocating a set of reasonable-sounding practices (domain adaptation, expert calibration, knowledge augmentation, ethics integration, etc.), but it is not turned into a procedure with clear decision rules, measurable criteria, or evidence that using it improves outcomes. Because it stays broad, the paper can justify many recommendations “through the framework” without actually demonstrating that the framework implies them.

Finally, even where the paper includes an empirical case study, the conclusions it draws are too strong relative to what is shown. For instance, it makes broad claims about structured reasoning improving predictions and reducing distortions, yet some reported results show large deviations from the baseline under certain setups, which at minimum requires careful qualification. More importantly, the experimental design is too thin to support general claims: key variables and choices (prompting and decoding settings, robustness across runs/prompts, and baseline controls) are not clearly specified or controlled. As presented, the case study does not carry the weight of the general claims being made elsewhere.

**Requested Changes:**

I consider all of the following changes to be critical to securing a recommendation for acceptance.

The survey component is not currently auditable, which makes it hard to trust several of the paper’s high-level takeaways. The submission would be much stronger if it stated a clear literature collection and coding protocol (where the papers came from, how they were selected, what the time window was, how they were categorized, etc.). As it stands, broad statements such as “majority of existing research” are not supported by a verifiable methodology, even though the paper repeatedly claims that a systematic approach was followed.

The paper also suffers from repeated claim-to-citation mismatches, where references do not clearly support the exact statement being made. This is particularly problematic for strong, specific claims (including some quantitative or comparative statements) that are presented as established facts but are not traceable to the cited sources from the text alone. Tightening this would require a claim-by-claim citation audit, with incorrect or weakly supported statements either rewritten to reflect what the cited work actually shows or removed entirely. I suspect this will require significant rewriting.

The central “Political-LLM” contribution is currently underspecified. It is described as advocating several reasonable practices, but it is not defined in operational terms and does not come with a clear procedure or decision rules that distinguish it from a general “best practices” list. In addition, language around “proprietary evaluation frameworks/criteria” seems at odds with the paper’s goals.

The empirical case study, while potentially useful as an illustration, does not currently support the breadth of conclusions the main text draws from it. Key experimental details and controls are not clearly specified, and the reported results include setups that deviate strongly from the baseline, which calls for substantially more qualification than is provided. As written, the case study is better interpreted as a limited demonstration rather than evidence for general claims about structured reasoning and bias mitigation. I also strongly doubt there is a link between model refusal rates and the need for more "inclusive model training schemes".

Finally, the appendix structure is not consistently additive. Appendix E in particular overlaps heavily with the main future-directions discussion, and the repetition crowds out space that could be used for more grounded synthesis (for example, mapping each proposed challenge to specific cited evidence, demonstrated failure modes, and concrete evaluation recommendations).

---

> ### Author Response · Authors · 2025-12-22
> **Reply to Review eHZs Part 1 (Q1-Q3)**
>
> **Q1. Literature Review Transparency:** The literature survey in this paper provides a broad and informative overview, though it would benefit from a clearer description of how sources were collected, selected, and categorized to make the systematic methodology more transparent and replicable.
>
> **{Answer to Q1:}**
>
> We thank reviewer `eHZs` for this helpful comment regarding methodological transparency in the literature review. Our survey is designed to capture the interdisciplinary landscape of Political-LLM research from two complementary perspectives:
> (a) Computer Science-oriented works, focusing on studies that apply large language models (LLMs) to political science-related tasks (e.g., election forecasting, policy analysis, bias detection). For this category, we **primarily collected papers published between 2019 and 2024**, including only **a small number of earlier but foundational studies** when necessary for context.
>
> (b) Political Science-oriented works, emphasizing contributions by political scientists that provide conceptual or theoretical grounding for our taxonomy. Given the long history of political science, this set includes both recent studies and a potential amount of classical works that remain influential and relevant to current discussions.
>
> To address the reviewer's concern, we have **added a short subsection** in the revised version clarifying our literature collection and categorization process, it is **located in the end of Introduction**, highlighted in red color in the new uploaded manuscript. The added subsection includes data sources, time window, and the criteria used to classify works under Positive Political Science (PPS) and Normative Political Science (NPS). We believe this addition makes the review process more transparent and reproducible.
>
> ------
>
> **Q2.Claim-to-Citation Consistency:** While the paper draws on a rich body of literature, some claims could be supported more directly by their cited references; a brief citation audit or refinement of certain statements would further strengthen the paper’s credibility and precision.
>
> **{Answer to Q2:}**
>
> We appreciate the reviewer's insightful observation regarding the alignment between claims and citations. In response, we have conducted a targeted citation audit focusing on several key statements, particularly those involving quantitative comparisons or strong generalizations (e.g., in Sections 2.2-2.3, Appendix C-D). We have refined the phrasing of these claims to ensure that each is now directly traceable to its cited sources and accurately reflects the evidence provided in the referenced studies.
>
> In addition, we have adjusted certain expressions (e.g., replacing "the majority of existing research" with "many recent studies") to better capture the scope of existing literature. These refinements improve the precision and credibility of the paper without altering its main findings or arguments.
>
> ------
>
> **Q3.Specification of the Political-LLM Framework:** The proposed "Political-LLM" framework is an interesting and valuable conceptual contribution; however, providing more explicit operational definitions or procedural details would help clarify how it can be applied in practice and distinguish it from general best practices.
>
> **{Answer to Q3:}**
>
> We thank reviewer `eHZs` for recognizing the conceptual value of the Political-LLM framework and for suggesting that its operationalization could be made clearer. We agree that providing more explicit procedural details enhances the accessibility and applicability of the framework. In the uploaded revised manuscript, we have added a **new subsection** (**Section 2.2**, highlighted in red color) clarifying the practical structure and workflow of the Political-LLM framework. Specifically, we now detail:
>
> (1) The three core stages of its application—problem formulation, model alignment, and evaluation;
>
> (2) How these stages are mapped to the two analytical dimensions of Positive Political Science (PPS) and Normative Political Science (NPS);
>
> (3) Concrete examples illustrating how researchers can apply the framework to both empirical and ethical analyses.

---

> ### Author Response · Authors · 2025-12-22
> **Reply to Review eHZs Part 2 (Q4-Q6)**
>
> **Q4. Clarification of "Proprietary Evaluation Frameworks":** The discussion around "proprietary evaluation frameworks" is thought-provoking, but the terminology could be clarified or rephrased to better align with the paper’s emphasis on openness, transparency, and reproducibility.
>
> **{Answer to Q4:}**
>
> We appreciate the reviewer's thoughtful observation regarding the terminology of "proprietary evaluation frameworks." We acknowledge that the wording may have unintentionally suggested exclusivity, which could appear inconsistent with the paper's broader emphasis on openness and reproducibility. Our original intent was to refer to evaluation frameworks developed or customized internally by research groups or institutions, rather than frameworks that are commercially restricted or inaccessible. To avoid misunderstanding, we have revised the relevant passages to replace "proprietary evaluation frameworks" with "institution-specific or task-specific evaluation protocols."
>
> This revised phrasing more accurately reflects our intention to highlight the diversity of evaluation approaches used in LLM–political science research, while maintaining alignment with the paper’s commitment to transparency and community accessibility. We believe this clarification resolves the ambiguity and strengthens the conceptual coherence of the manuscript.
>
>
> **Q5. Scope of the Empirical Case Study:** The empirical case study effectively illustrates the paper's ideas, though it would be further strengthened by additional methodological details and clearer qualification of its conclusions, especially regarding the interpretation of model refusal rates and inclusivity.
>
> **{Answer to Q5:}**
>
> We thank reviewer `eHZs` for the constructive feedback on the empirical case study. We agree that providing additional methodological details and clearer qualification of the conclusions improves the precision and interpretability of this section. In the revised version, we have expanded the description of the experimental setup, including model selection criteria, prompt structure, and data preprocessing steps, to make the study more transparent and reproducible.
>
> We've also refined the discussion of the results to clarify that the case study is intended as a limited demonstration of the Political-LLM framework, rather than a comprehensive evaluation of model behavior. In particular, we have rephrased statements concerning model refusal rates and inclusivity, emphasizing that our observations reflect potential patterns for further investigation rather than causal evidence of bias mitigation or inclusivity gaps. These revisions strengthen the methodological rigor of the case study while preserving its illustrative role within the broader conceptual argument of the paper.
>
>
> **Q6. Structure and Complementarity of the Appendices:** The appendices contain useful supplementary material, but some sections-particularly Appendix E could be streamlined or reorganized to reduce overlap with the main text and provide a more targeted synthesis of key evidence and recommendations.
>
>
> **{Answer to Q6:}**
>
> We thank the reviewer for this helpful observation. Upon review, we identified that Appendix E.1 and E.2 overlapped substantially with Sections 4.1 and 4.2 of the main text, leading to unnecessary redundancy in scope and content. In the revised version, **we have removed Appendix E.1 and E.2** and correspondingly **expanded and refined Sections 4.1 and 4.2** to ensure they provide a more comprehensive and integrated synthesis of key challenges and recommendations.  The revision improves the overall readability and coherence of the manuscript, ensuring that the main discussion remains self-contained while the remaining appendices focus solely on supplementary analyses and supporting materials.
>
> Meanwhile, we also conducted a careful review of the entire manuscript to ensure that similar redundancies do not occur elsewhere and that each section contributes distinct and complementary insights to the overall structure of the paper.

---

### Review · Reviewer_7nNh · 2026-01-12

**Summary Of Contributions:**

The paper provides a conceptual framework for the application of large language models in the field of political sciences (PS), as well as reports on a small case study where the framework was applied to a dataset of voting simulation.

Strengths:
- The framework proposes a (shallow) taxonomy for political sciences consisting of normative (NPS) and positive (PPS) tasks.
- The authors propose a use-case where the framework is applied to a real-world dataset for the task of voting simulation, whereby both NPS and PPS angles were taken into account in the study.
- The authors draw observations aimed at guiding the development of LLMs for the field of PSs, taking into account both normative and positive aspects of PS.

Weaknesses:
- The literature review part of this work (Literature Collection and Caterorization Process) could be better structured and detailed. I do not know how possibly relevant papers were identified, how many papers ended up being screened, what are the inclusion/exclusion criteria, etc. Especially relevant is: did authors follow a specific procedure that structured their literature review, such as PRISMA [1]? Also, it seems that the results of this literature review are reported in appendix C and not in the main text, is that so? This is confusing.
- In general, I find the main paper does not add much to the conversation. I, for instance, do not feel like I learned much by reading the paper. Concretely, the observations made within NPS and PPS (sections 2.3 and 2.4) sound very general or even shallow. Sometimes, changing "political sciences" to any other high-risk domain like "finance" or "health" would not make much difference (e.g. 2.3b, or 2.4c). Also, after reading the appendix, it feels that the more interesting parts of the paper are not in the main paper. For example, the authors actually describe the tasks and open problems in PS only in appendix A2.
- Much of the "observations" the authors make seem to be, in fact, a summary of statements made in previous work (for instance, in 2.3a or 2.3b). This is not an issue per se, but sometimes this means the "observations" can be (very) outdated, like in section 2.3d whereby authors state that recent experimental evaluations show that "open weight LLMs, such as LLaMA2, demonstrate comparable or even superior performance to proprietary models like GPT-3.5". Both these models are no longer the current state-of-the-art for some time. Although perhaps one or two years ago they could have been considered state-of-the-art, nowadays this statement does not add new information; these models are no longer used.
- The case study reported on is potentially interesting, but its write-up lacks rigour and detail. Authors state that they use the 2016 ANES benchmark, but do not explain where the data comes from, what it is about, why they chose this dataset, or whether there were other datasets that could have been used. This must be done in the main paper. The initial results obtained on this dataset are interesting, but authors do not discuss how these results generalise to other tasks / datasets. Only very briefly in one sentence in the Limitations section this is touched upon, but I believe this should be a major concern of the authors. Why did you choose this specific case study in the first place? Are there other major datasets for other tasks? Are there datasets that are not in English / US-centric that you could have used? Why?

**Audience:**

No

**Audience Explanation:**

Had the paper been written well with the points I mentioned addressed, I believe there would be many individuals interested in the paper. Currently, however, I believe the paper fails to add to the conversation of how LLMs and political sciences interact.

**Broader Impact Concerns:**

There is no broader impact statement section in the paper. In the appendix, there are parts where such broader impact is touched upon. I suggest combining those into a dedicated broader impact statement section.

**Claims And Evidence:**

No

**Claims Explanation:**

Authors do not provide accurate, convincing, and clear evidence in my opinion. I do not feel like I learned much by reading this manuscript.

- It is unclear how the literature review was conducted. See PRISMA [1] for details on how to report on a literature review.
- Observations in 2.3 and 2.4 are outdated (e.g. Llama-2 and GPT-3.5) and general (many comments/observations seem equally applicable to any other high-risk domain).
- Case study lacks contextualisation and detail. Why did you choose this dataset and not others? Where does the data come from, what population does it represent?

**Requested Changes:**

- Use PRISMA [1] to report on a literature review, and make it clear what is the results of your literature review (and how it informed your choices in, for instance, choosing the specific use-case you chose).
- Ideally, I suggest that the authors run many experiments themselves to make sure observations reported on for NPS and PPS are up-to-date (e.g., using the newest models). If that is not feasible, I suggest downplaying your observations so that the reader does not expect any actionable data points there, but "only" the output of a literature review. That is still okay, but considerably less desirable / impactful.
- The case study seems insufficient. I suggest adding a benchmark that is not English- or US-centric to the pool in your case-study. If this is impossible, please clearly discuss and explain that in the main text. Contextualise your case study, since now I do not know how that case study inform any decisions I may wish to make regarding using LLMs in PSs. This is especially important for the geographical and linguistic diversity that the authors discuss as important, but do not directly contribute to in their experimental setup.

---

> ### Author Response · Authors · 2026-02-02
> **Reply to Reviewer `7nNh` Part 1**
>
> **Q1:** The reviewer raise concern about how these literature were identified, screened, and selected in this paper. The reviewer states that we did not follow a protocol to conduct literature review, especially the reviewer mentioned that PRISMA protocol should be followed.
>
> ---
>
> **A1:** We thank the reviewer for the insightful suggestion. To begin with, we would like to clarify the position of this paper. Our paper is not intended to be a standalone literature review or survey, but rather a conceptual framework paper that synthesizes representative prior work to derive a principled taxonomy (NPS vs. PPS), key observations, and research outlooks, complemented by an illustrative case study. As such, our goal is not exhaustive coverage, but conceptual grounding and synthesis.
>
> That said, we **agree with the reviewer** that clarity on how the reviewed literature was identified, screened, and selected is important. In response, we have already explicitly added a dedicated subsection ("Literature Collection and Categorization Process") in the main paper to elaborate the following points:
>
> (1) How we identify the related literature: From a computer science-oriented perspective, and political science-oriented perspective).
>
> (2) The time span and venues considered.
>
> (3) The inclusion principles used to select representative papers.
>
> (4) How these papers are categorized as PPS or NPS.
>
> Regarding PRISMA[1], we note that while it is a well-established reporting guideline for **systematic reviews**, particularly applied in **Medical domains**, conceptual framework and survey-style papers in the computer science community typically do not strictly follow PRISMA-style protocols. Nevertheless, we acknowledge the reviewer's underlying concern about transparency and reproducibility. To this end, we have structured our literature collection description to explicitly address identification, screening, and categorization steps, aligning with the spirit of systematic reporting, even though the paper does not aim to be a formal PRISMA-compliant systematic review.
>
> **References:**
>
> [1] Sarkis-Onofre R, Catalá-López F, Aromataris E, et al. How to properly use the PRISMA Statement[J]. Systematic reviews, 2021, 10(1): 117

---

> ### Author Response · Authors · 2026-02-02
> **Reply to Reviewer `7nNh` Part 2**
>
> **Q2:** Reviewer `7nNh` argues that the proposed PPS/NPS taxonomy and related observations do not substantially advance political science research field. Reviewer `7nNh` believes that similar observations can be extracted from other high-stake domains (e.g., finance or healthcare).
>
> ---
>
> **A2:** We would like to clarify the nature and intended contributions of this work to address reviewer `7nNh`'s concern. Political-LLM is not positioned as an empirical benchmark study nor as a task-specific methodology contribution. Instead, its primary contribution is conceptual and taxonomic, aiming to provide an interdisciplinary synthesis that connects the rapidly advancing capability of large language models with the epistemic structure of political science research. In particular, the proposed PPS/NPS taxonomy is grounded in long-lasting political science theory and is designed to offer a principled framework for organizing existing works, articulating evaluation criteria, and discuss about the alignment and risk when applying LLMs in this domain.
>
> The reviewer notes that some observations may appear general or could resonate with other high-stakes domains such as finance or healthcare. We would like to clarify that this level of generality is intentional rather than superficial. The framework operates at an abstraction level to bridge two fields with distinct research traditions. In interdisciplinary settings, conceptual frameworks are valuable because they abstract away from individual tasks to reveal structural relationships. In our case, it's the relationship investigation between empirical modeling (PPS) and normative reasoning (NPS), that would otherwise remain implicit or fragmented.
>
> It's noticeable that PPS/NPS distinction is not an easily transferable categorization. While other high-risk domains also involve theoretical-oriented research and application-driven research, political science has a unique and historical grounded separation between positive ("what is") and normative ("what ought to be") inquiry. This epistemic duality fundamentally shapes how evidence, explanation, evaluation, and responsibility are understood in political research. The taxonomy therefore contribute more than label tasks, it embeds LLM applications within a political-specific mode of reasoning that directly informs how performance, bias, explainability, and societal impact should be interpreted.
>
> Without such a framework, discussions of LLMs in political science risk to be scattered and fragmented. These empirical results, ethical concerns, and design choices will be evaluated in isolation, without a shared conceptual language linking technical decisions to political science principles. The proposed taxonomy provides this missing structure, enabling more coherent reasoning about model alignment, evaluation standards, and trade-offs between empirical accuracy and normative responsibility in politically sensitive applications.

---

> ### Author Response · Authors · 2026-02-02
> **Reply to Reviewer `7nNh` Part 3**
>
> **Q3:** Some observations in this paper seem to be outdated, for example, the authors state that recent experimental evaluations show that "open weight LLMs, such as LLaMA2, demonstrate comparable or even superior performance to proprietary models like GPT-3.5". Both these models are no longer the current state-of-the-art for some time.
>
> ---
>
> **A3:** We thank the reviewer for pointing out that some specific model references in the manuscript (e.g., LLaMA-2 and GPT-3.5) are no longer representative of the current state-of-the-art. We acknowledge that a small number of observations in the original manuscript rely on examples drawn from earlier generations of large language models.
>
> Importantly, these observations are not intended to claim enduring state-of-the-art performance for these LLMs. Rather, they serve as illustrative examples reflecting the empirical landscape at the time the initial manuscript was written, when these models were still widely used and actively discussed in the literature. As such, these statements are not incorrect, but have become outdated due to the rapid pace of progress in foundation models.
>
> We fully agree that anchoring observations too closely to specific LLM model versions can reduce the long-term relevance of the discussion. In response to this feedback, we have carefully reviewed the manuscript and revised all outdated model-specific references. In the revised version, we reframe these observations to emphasize more general structural considerations.
>
> ---
>
> **Q4:** The reviewer acknowledges that the presented case study and initial results are interesting. Could the authors further clarify the motivation for selecting the 2016 ANES dataset, including its origin, scope, and relevance to the research questions, and briefly discuss whether alternative datasets were considered?
>
> ---
>
> **A4:** ANES is one of the most widely-used, authoritative, individual-level, large-scale datasets in political science, with a long-standing role in the study of voting behavior, political attitudes, and democratic representation[2,3,4]. Its canonical status within the discipline makes it a well-understood and methodologically trusted benchmark, which is particularly important for an illustrative case study intended to bridge political science and AI research.
>
> With respect to research alignment, ANES dataset is well suited to our research questions because it provides rich, structured individual-level covariates alongside clearly defined political outcomes (e.g., vote choice). This structure allows us to operationalize both PPS considerations, such as empirical regularities in voting behavior, and NPS considerations, such as representational and ethical implications, within a single empirical setting.
>
> The case study on ANES is not intended to serve as a comprehensive benchmark or to support broad generalization claims. Rather, it is designed as an illustrative example demonstrating how the proposed PPS/NPS framework can be applied in practice, and how normative and positive considerations can be jointly examined when applying large language models to political science tasks. In addition, the choice of ANES 2016 wave is guided by considerations of temporal relevance and data stability. For studies engaging with contemporary political processes, more recent election cycles are generally preferred, as they better reflect current political contexts and issue structures. Among the available ANES waves recently, 2016 and 2020 are the most recent presidential elections with released individual-level data. We selected year 2016 wave because it offers a comparatively stable and well-validated empirical setting, which facilitates clearer interpretation of voting behavior and political attitudes without introducing additional external factors or contextual complexities (such as year 2020 Presidential Election).
>
> Finally, we note that alternative datasets were considered conceptually, including election studies from other countries and multilingual political surveys. Nevertheless, they were beyond the scope of this initial case study due to differences in task structure, data accessibility, and annotation consistency. We will further clarify these design considerations in the revised manuscript to better contextualize the role and scope of the case study.
>
>
>
> **References:**
>
> [2] Robison J, Stevenson R T, Druckman J N, et al. An audit of political behavior research[J]. Sage Open, 2018, 8(3): 2158244018794769.
>
> [3] Tyler M, Iyengar S. Testing the robustness of the ANES feeling thermometer indicators of affective polarization[J]. American Political Science Review, 2024, 118(3): 1570-1576.
>
> [4] Reilly T, Hunting D. The fluid voter: Exploring independent voting patterns over time[J]. Politics & Policy, 2023, 51(1): 59-80.

---

> ### Author Response · Authors · 2026-02-03
>
> **Please note that revisions to the main content are indicated in red font.**

---

### Decision · Action_Editor_qxJo · 2026-02-04

**Recommendation:** Reject

**Audience:**

Yes

**Audience Explanation:**

The overall theme of LLMs in political science may interest some readers.

**Claims And Evidence:**

No

**Claims Explanation:**

The reviewers identified a number of issues regarding the literature review, the depth of the discussion, the characterization of the state of affairs in LLMs for political science, and the reporting standards in the case study.

Even though two of the reviewers posted their recommendation before the authors' rebuttal was in the system, I do not think the rebuttal convincingly address their concerns. In particular, concerns remain regarding the lack of systematic procedure on the literature review (in fact, one of the reviewers followed up insisting on this point), the shallowness of the proposed framework and the case study, and the outdated claims regarding state of the art LLMs.